# Natural product-mediated reaction hijacking mechanism validates *Plasmodium* aspartyl-tRNA synthetase as an antimalarial drug target

Nutpakal Ketprasit[1]◉, Chia-Wei Tai[1]◉, Vivek Kumar Sharma[2]◉, Yogavel Manickam[2], Yogesh Khandokar[3], Xi Ye[1], Con Dogovski[1], David H. Hilko[4,5], Craig J. Morton[6], Anne-Sophie C. Braun[4,5], Michael G. Leeming[7], Bagale Siddharam[8], Gerald J. Shami[1], Pushpangadan Indira Pradeepkumar[8], Santosh Panjikar[3,9], Sally-Ann Poulsen[4,5], Michael D. W. Griffin[1]*‡, Amit Sharma[2]*‡, Leann Tilley[1]*‡, Stanley C. Xie[1,10]*‡

1 Department of Biochemistry and Pharmacology, Bio21 Molecular Science and Biotechnology Institute, The University of Melbourne, Melbourne, Australia, 2 Molecular Medicine - Structural Parasitology Group, International Centre for Genetic Engineering and Biotechnology, New Delhi, India, 3 Australian Synchrotron, ANSTO, Clayton, Australia, 4 Institute for Biomedicine and Glycomics, Griffith University, Nathan, Australia, 5 School of Environment and Science, Griffith University, Nathan, Australia, 6 Biomedical Manufacturing Program, CSIRO, Clayton South, Australia, 7 Melbourne Mass Spectrometry and Proteomics Facility, Bio21 Molecular Science and Biotechnology Institute, The University of Melbourne, Melbourne, Australia, 8 Department of Chemistry, Indian Institute of Technology Bombay, Mumbai, India, 9 Department of Biochemistry and Molecular Biology, Monash University, Clayton, Australia, 10 Drug Delivery Disposition and Dynamics, Monash Institute of Pharmaceutical Sciences, Monash University, Parkville, Australia

◉ These authors contributed equally to this work.
‡ These authors are jointly supervised this work.
* amit.icgeb@gmail.com (AS); mgriffin@unimelb.edu.au (MDWG); ltilley@unimelb.edu.au (LT); stanley.xie@monash.edu (SCX)

## Abstract

Malaria poses an enormous threat to human health. With ever-increasing resistance to currently deployed antimalarials, new targets and starting point compounds with novel mechanisms of action need to be identified. Here, we explore the antimalarial activity of the *Streptomyces sp* natural product, 5′-O-sulfamoyl-2-chloroadenosine (dealanylascamycin, DACM) and compare it with the synthetic adenosine monophosphate (AMP) mimic, 5-O-sulfamoyladenosine (AMS). These nucleoside sulfamates exhibit potent inhibition of *P. falciparum* growth with an efficacy comparable to that of the current front-line antimalarial, dihydroartemisinin. Exposure of *P. falciparum* to DACM leads to inhibition of protein translation, driven by eIF2α phosphorylation. We show that DACM targets multiple aminoacyl-tRNA synthetases (aaRSs), including the cytoplasmic aspartyl tRNA synthetase (AspRS). The mechanism involves hijacking of the reaction product, leading to the formation of a tightly bound inhibitory amino acid-sulfamate conjugate. We show that recombinant *P. falciparum* and *P. vivax* AspRS are susceptible to hijacking by DACM and AMS, generating Asp-DACM and Asp-AMS adducts that stabilize these proteins. By contrast, human AspRS appears less susceptible to hijacking. X-ray crystallography reveals that apo *P. vivax* AspRS exhibits

**Data availability statement:** All relevant data are within the manuscript and its Supporting Information files. The following structures have been deposited in the PDB: PvAspRS-apo (9M5M); PvAspRS:Asp-AMP (9M5N), PvAspRS:Asp-AMS (9M5O); PvAspRS:Asp-DACM (9NPJ).

**Funding:** We would like to acknowledge funding from the Australian National Health and Medical Research Council (APP2019492 to LT and SCX) and the Australian Research Council (DP220102618 to S.-A.P; and DE230101173 to SCX). This research was supported by AINSE Ltd. Postgraduate Research Award (PGRA to NK). We would like to acknowledge funding from the Indian Council of Medical Research (CAR-2024-01-000140 to AS and PPI), the Anusandhan National Research Foundation (CRG/2023/004351 to AS and PPI), the Department of Biotechnology (PR32713) and a J C Bose National Fellowship (SB/S2/JCB-41/2013 to AS) from the Science and Engineering Research Board (SERB) of Department of Science and Technology (DST), Government of India. Funders did not play any role in the study design, data collection and analysis, decision to publish, or preparation of the manuscript.

**Competing interests:** The authors have declared that no competing interests exist.

a stabilized flipping loop over the active site that is poised to bind substrates. By contrast, human AspRS exhibits disorder in an extended region around the flexible flipping loop as well as in a loop in motif II. These structural differences may underpin the decreased susceptibility of human AspRS to reaction-hijacking by DACM and AMS. Our work reveals *Plasmodium* AspRS as a promising antimalarial target and highlights structural features that underpin differences in the susceptibility of aaRSs to reaction hijacking inhibition.

## Author summary

Every year, at least 200 million new malaria infections are diagnosed, causing more than 500,000 deaths, mostly in Africa. Increasing levels of resistance to current antimalarials were exacerbated by disruptions to health services during the COVID pandemic. Breakthrough drugs with new modes of action are needed. *Streptomyces*, a bacterium mainly found in decaying vegetation, exhibits a complex secondary metabolism that has provided two-thirds of the clinically used antibiotics of natural origin. This work characterizes a *Streptomyces sp* natural product, dealanylascamycin (DACM), that kills malaria parasites in culture with an efficacy comparable to that of the current front-line antimalarial, dihydroartemisinin. DACM works via an unusual reaction-hijacking mechanism. Several of the parasite's aminoacyl tRNA synthetases are induced to link DACM to an amino acid in the active site to form a tight-binding adduct that disrupts the malaria parasite's protein translation machinery. We structurally and biochemically characterized the DACM-inhibited malaria parasite aspartate tRNA synthetases to understand why it is more susceptible than the equivalent human enzyme.

## Introduction

In 2023, *Plasmodium falciparum* caused 263 million cases of malaria, resulting in 597,000 deaths, mostly of African children [1]. Resistance of the mosquito vectors to pyrethroid insecticides [2] and widespread resistance of parasites to currently used therapies [3] was compounded by disruptions to prevention strategies during the COVID-19 pandemic [4]. The recent emergence of artemisinin resistance-conferring K13 mutations in Africa [5,6] is of particular concern. Thus, there is a need to explore new avenues for the development of antimalarial compounds with novel mechanisms of action.

Protein translation relies on aminoacyl-tRNA synthetases (aaRSs) to charge tRNAs with their cognate amino acids [7]. The essentiality of this process means aaRSs represent promising potential targets for antimalarials [8,9]. For example, the natural product-derived mupirocin, an isoleucyl-tRNA synthetase (IleRS) inhibitor, is widely used as a topical antibiotic [10], while halofuginone, a prolyl-tRNA synthetase (ProRS) inhibitor, is used to prevent coccidiosis in poultry [11].

Nucleoside sulfamates are chemical compounds that use an unusual reaction-hijacking mechanism to inhibit enzyme function, resulting in new clinical candidates (*e.g.,* Pevonedostat, TAK-243 and TAK-981 [12–14]). For many years, the reaction hijacking mechanism was thought to be applicable only to ubiquitin-activating (E1) enzymes. Recently, two different classes of AMP-mimicking nucleoside sulfamates/ sulfonamides have been explored that inhibit *P. falciparum* aaRSs via a reaction-hijacking mechanism. The pyrazolo-pyrimidine sulfamates, ML901 [15] and ML471 [16], and the amino-thieno pyrimidine benzene sulfonamide, OSM-S-106 [17], were shown to target *P. falciparum* tyrosyl-tRNA synthetase (*Pf*TyrRS) and asparaginyl-tRNA synthetase (*Pf*AsnRS), respectively, providing potent and specific anti-plasmodial activity. The target enzymes catalyze coupling of the nucleoside sulfamate/ sulfonamide to the amino acid substrate, forming a tight binding uncleavable adduct [15–17].

While ML901, ML471 and OSM-S-106 target specific aaRSs, adenosine 5′-sulfamate (AMS), which is a direct bio-isostere of adenosine monophosphate (AMP), was shown to be a broadly reactive hijacking inhibitor of both Class I and Class II *P. falciparum* aaRSs [15]. This opens the possibility of designing bespoke nucleoside sulfamates with tunable specificity.

To explore the range of potential hijackable targets, here we investigated a natural analogue of AMS, known as dealanylascamycin (5′-*O*-sulfamoyl-2-chloroadenosine; DACM) (Fig 1A). DACM is a natural product nucleoside sulfamate from a soil-dwelling *Streptomyces* bacterium. Earlier studies showed that it exhibits broad-spectrum antibacterial and herbicidal activities [18,19]. The mechanism of inhibition was not elucidated, but inhibition of bacterial protein translation was reported [20–23]. We hypothesized that DACM is likely a reaction-hijacking inhibitor.

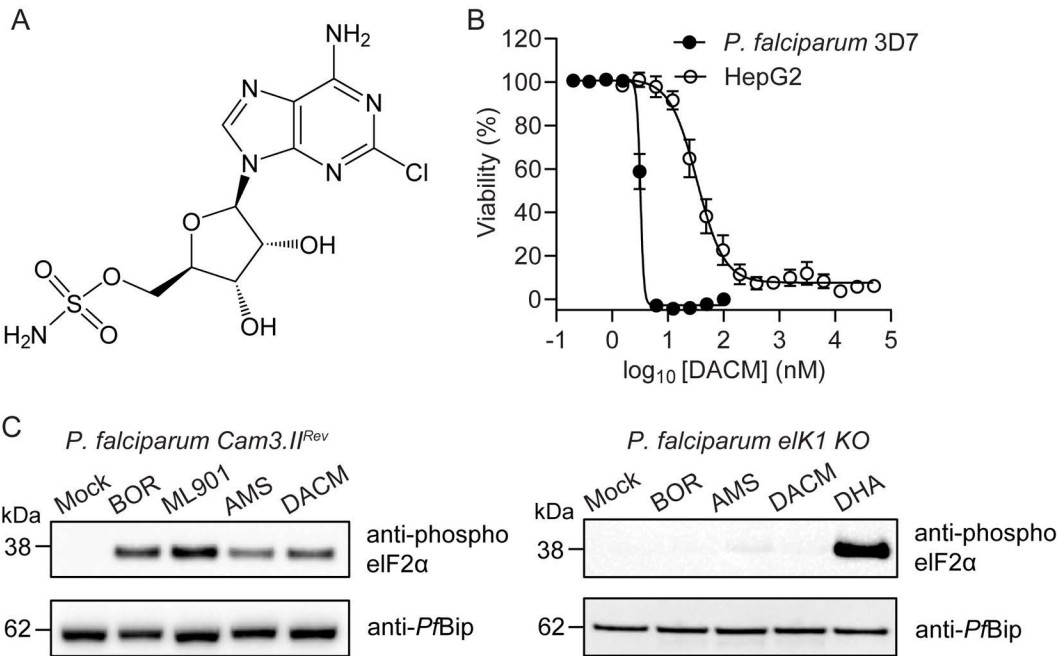

**Fig 1. DACM inhibits *P. falciparum* growth and induces the amino acid starvation response.** (A) Structure of DACM. (B) Sorbitol-synchronized ring stage parasites were subjected to a 72-h drug sensitivity assay with DACM (black circles). Data represent five independent experiments, each performed in duplicate. Cytotoxicity of DACM (white circles) against the HepG2 mammalian cell line, subjected to a 48-h exposure. Data represent four independent experiments each performed in triplicate. Error bars indicate SEM. (C) Trophozoite-stage *P. falciparum* Cam3.II^rev or *P. falciparum* eIK1 knockout (KO) cultures were exposed to 0.2 μM borrelidin (BOR), 1 μM ML901, 1 μM AMS, 1 μM DACM or 1 μM DHA for 3 h. Parasite extracts were analysed by Western blot analysis for phosphorylated eIFα. *Pf*Bip is the loading control. The blot is representative of three independent experiments.

Here we show that DACM inhibits growth of *P. falciparum* cultures by inhibiting protein translation. Targeted identification of DACM adducts confirms multiple aaRS targets in *P. falciparum*, including cytoplasmic aspartyl tRNA synthetase (AspRS). Via thermal stabilization, targeted mass spectrometry and structural studies of recombinant AspRS from *P. falciparum* and *P. vivax*, we reveal the molecular basis for differential susceptibilities between host and parasite enzymes. This work thus provides a new focus for antimalarial drug development centered on AspRS.

## Results

### Synthesis of DACM, Asp-DACM and Asp-AMS

The synthesis of DACM has been described previously [24,25]. The synthesis of 5′-O-(N-L-aspartate)-sulfamoyl-adenosine (**Asp-AMS;** Scheme 2 in S1 Text) was performed as detailed in S1 Text. 2-chloro- 5′-O-[N-(L-aspartyl)-sulfamoyl] adenosine (Asp-DACM) was prepared by adapting literature methods [26,27]. Briefly, the bis-protected *N*-(Boc)-*tert*-butyl aspartic acid was converted to the *N*-hydroxysuccinate ester (**1**) with *N*-hydroxysuccinimide and *N,N'*-dicyclohexylcarbodiimide (DCC). Next, the aspartate *N*-hydroxysuccinimide ester (**1**) was coupled directly with the 5′-O-sulfamoylated isopropylidene protected adenosine (**2**) in the presence of 1,8-diazabicyclo[5.4.0]undec-7-ene (DBU) in *N,N*-dimethylformamide (DMF) to yield the triply protected *N*-aminoacylated sulfamoyl adenosine derivative (**3**). Lastly, complete global deprotection of **3** with trifluoroacetic acid (TFA) in water and tetrahydrofuran (THF) gave the desired 2-chloro-5′-O-[N-(ʟ-aspartyl)-sulfamoyl] adenosine derivative, **Asp-DACM**, in high purity after precipitation from acetonitrile (MeCN) and triethylamine (Et₃N) (Scheme 1 in S1 Text).

### Effect of DACM on *P. falciparum* growth, stress response and protein translation, and toxicity to mammalian cells

Synchronized ring stage parasites (3D7 strain [28]) were exposed to increasing concentrations of DACM, and parasite viability was assessed in the next cycle by flow cytometry [29]. DACM is a potent inhibitor of the growth of *P. falciparum* cultures (Fig 1B; $IC_{50\_72h} = 3.3 \pm 0.1$ nM (n = 5)) with an efficacy similar to the previously reported *Pf*TyrRS inhibitor ML471 [16], and to the current front-line drug, dihydroxyartemisinin (DHA) [17]. It should be noted that DACM also exhibits significant toxicity against the mammalian cell line, HepG2, with an $IC_{50\_48h}$ value of $47 \pm 10$ nM (n = 4) (Fig 1B). Adenosine 5′-sulfamate (AMS, S1A Fig), a structurally related compound [15], also exhibits potent activity against *P. falciparum* cultures (S1B Fig; $IC_{50\_72h} = 3.8 \pm 0.2$ (n = 5)). AMS inhibits the growth of HepG2 with a $IC_{50\_48h}$ value of $53 \pm 2$ nM (n = 5) (S1B Fig).

Accumulation of uncharged tRNAs triggers the amino acid starvation stress response. Using established methods [17], we showed that eukaryotic initiation factor 2α (eIF2α) is phosphorylated upon exposure of *P. falciparum* cultures to DACM and AMS (Fig 1C, left panel). A similar response was observed for ML901, a known reaction hijacking inhibitor of *Pf*TyrRS, as well as the conventional tRNA synthetase inhibitor, borrelidin (Fig 1C, left panel). By contrast, in transfectants in which the eukaryotic translation initiation factor 2-alpha kinase 1 (eIK1) has been deleted [30], DACM and other aaRS inhibitors did not cause eIF2α phosphorylation (Fig 1C, right panel). As previously reported [31], exposure of the eIK1 knockout to dihydroxyartemisinin (DHA) still induces eIF2α phosphorylation (Fig 1C, right panel).

The aaRSs charge tRNAs with amino acids to drive protein synthesis. We monitored protein translation in *P. falciparum* by following the incorporation of an alkyne analogue of puromycin (OPP), followed by attachment of a clickable fluorophore [32,33]. Upon treatment of trophozoite stage parasites with DACM or AMS, protein translation was inhibited with $IC_{50\_3h}$ values of $32 \pm 1$ and $27 \pm 6$ nM, respectively (Fig 2A and 2B). These values correlate well with the $IC_{50\_3h}$ values observed for parasite killing of $22.6 \pm 0.2$ and $18 \pm 4$ nM, respectively (Fig 2A and 2B). These data are consistent with inhibition of protein translation being directly linked to parasite killing. An equivalent 4-h exposure to the folate pathway inhibitor, WR99210, kills parasites without affecting protein translation in the period monitored (Fig 2C), while a ribosome-directed protein translation inhibitor, cycloheximide, inhibits translation, but this short exposure results in limited killing (Fig 2D).

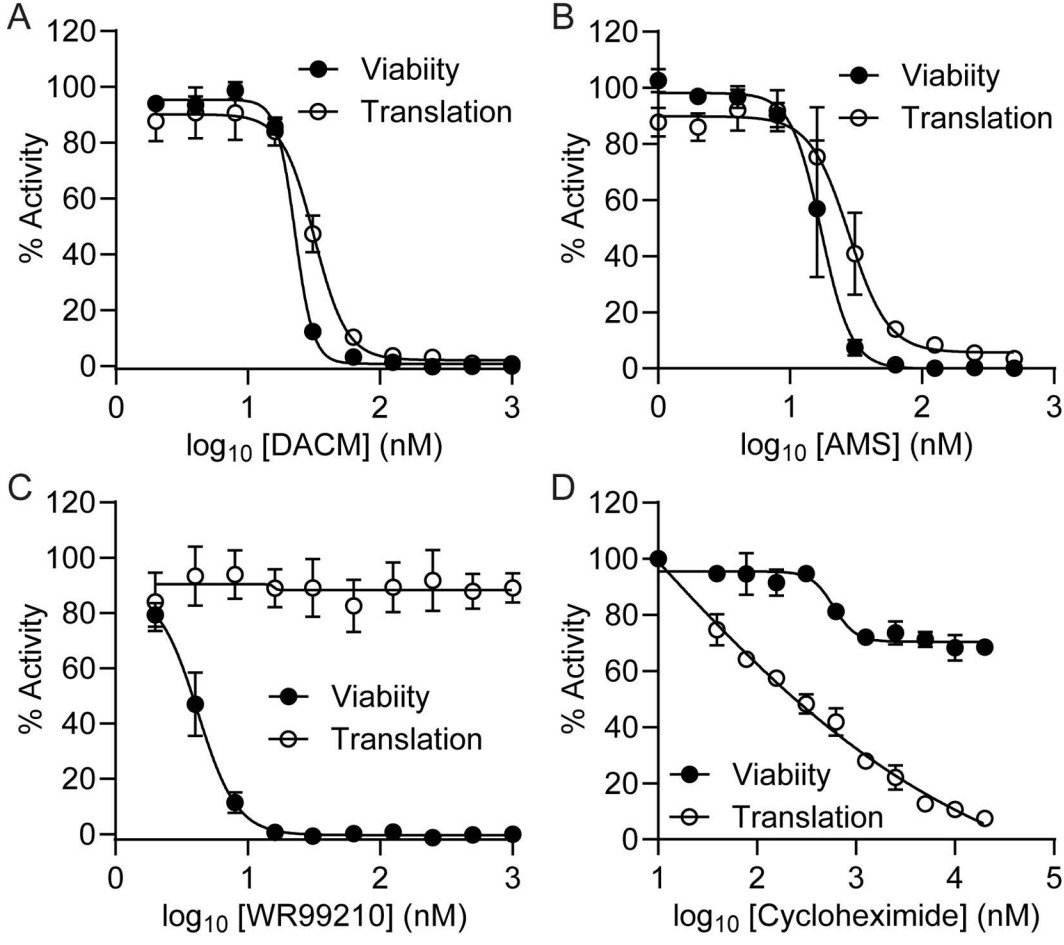

**Fig 2. DACM treatment inhibits protein translation in *P. falciparum* cultures.** Late-stage *P. falciparum* (Cam3.II^rev) infected RBCs were exposed to the inhibitors for 6 h with the incorporation of OPP during the final 2 h. For parasite viability, compounds were washed out after the 6-h exposure, and viability was assessed in the next cycle. (A) DACM, (B) AMS, (C) WR99210, (D) Cycloheximide.

## DACM targets multiple aaRSs in *P. falciparum*, including *Pf*AspRS but not *Pf*TyrRS

Reaction hijacking nucleoside sulfamates attack the activated oxy-ester bonds of the enzyme-bound aminoacyl tRNA to form tight-binding adducts (Fig 3A). We used targeted mass spectrometry to search for potential conjugates in *P. falciparum*-infected red blood cells (RBCs) that had been treated with DACM (10 μM, for 3 h). Following Folch extraction of lysates, the aqueous phase was subjected to LCMS and the anticipated masses for the 20 amino acid conjugates were interrogated. Signals were observed with precursor *m/z* values within 5 ppm of theoretical values calculated for DACM adducts of Asn (most abundant), as well as Asp, Thr, Ser and Lys, while weaker signals were observed for His and Phe (Figs 3B, 3C and S2). The identity of the Asp-DACM adduct was confirmed using a synthetic standard, which exhibited essentially the same retention time, precursor ion *m/z* value and MS/MS fragmentation spectrum as the species generated by *P. falciparum* (Fig 3B-D). No corresponding peaks were detected in control samples not exposed to DACM. The data indicate that at least *Pf*AsnRS, *Pf*AspRS, *Pf*ThrRS, *Pf*SerRS, *Pf*LysRS, *Pf*HisRS and *Pf*PheRS are susceptible to reaction hijacking by DACM.

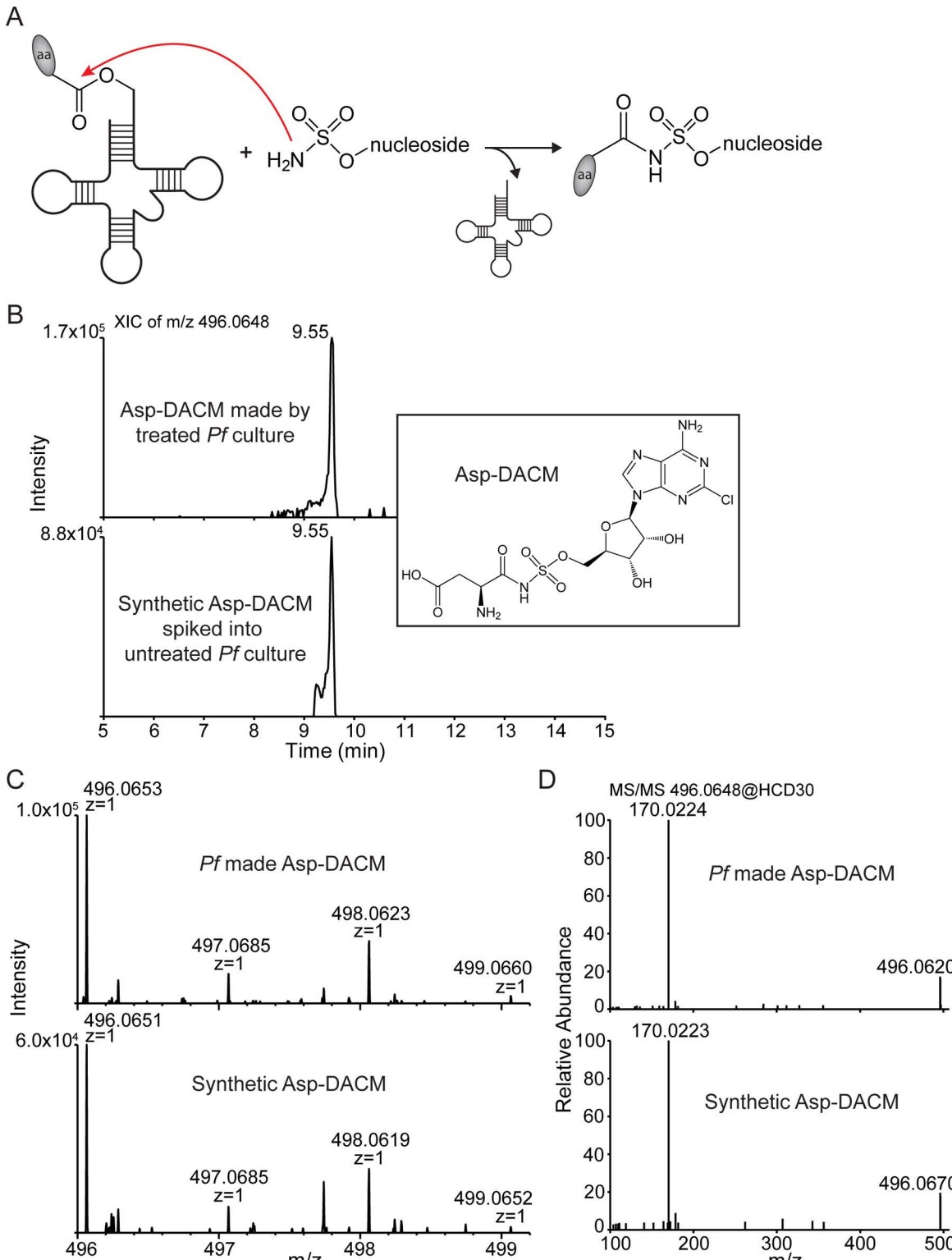

**Fig 3. Targeted mass spectrometry identifies Asp-DACM conjugates in *P. falciparum.*** (A) aaRSs catalyze nucleoside sulfamate attack on activated amino acids (aa) to form an amino acid adduct. (B-D). Late trophozoite stage *P. falciparum* 3D7 cultures were exposed to 10 µM DACM for 3 h. Parasite extracts were subjected to LC-MS/MS to search for DACM-amino acid conjugates. (B) Extracted ion chromatogram (XIC) of Asp-DACM ($m/z = 496.0648$)

from *P. falciparum* extracts and synthetic Asp-DACM standard spiked into untreated parasite lysate. Inset: Structure of Asp-DACM. (C,D) Mass spectra (C) and MS/MS fragmentation spectra (D) of Asp-DACM produced by *P. falciparum* extracts (top panels) and synthetic Asp-DACM standard (bottom panels).

### *Pf*TyrRS is less susceptible to hijacking by DACM than by AMS

The lack of Tyr-DACM adduct signal suggested that *Pf*TyrRS is not a target of DACM. This contrasts with AMS, for which Tyr-AMS was the most easily detected product in mass spectrometry data [15]. We generated recombinant *Pf*TyrRS as described previously [17]. We used differential scanning fluorimetry (DSF) [17] to monitor changes in the thermal stability of recombinant *Pf*TyrRS upon incubation with AMS and DACM. Under the conditions examined, AMS exposure caused marked stabilization of *Pf*TyrRS (a 10°C shift), while DACM did not (S3A Fig and S1 Table), suggesting weaker hijacking activity. To understand why *Pf*TyrRS might be less susceptible to hijacking by DACM, we carried out protein–ligand docking using the Surflex fragment matching strategy [34]. Unconstrained docking always resulted in unrealistic orientations for Tyr-DACM, so the position of the Tyr fragment of Tyr-AMP-bound *Pf*TyrRS (PDB: 7ROR) was used to constrain the position of the Tyr-moiety in Tyr-AMP, Tyr-AMS and Tyr-DACM, which were docked into the active site *in silico* (without protein flexibility). Tyr-DACM gives the least favorable docking score (S2 Table), due to a clash of the Cl atom with surrounding residues (S3B Fig).

### Generation of recombinant AspRSs

*Pf*AspRS was chosen for further analysis as it has not been studied as a reaction-hijacking target previously. Alignment of the AspRS amino acid sequences from *P. falciparum*, *P. vivax, Saccharomyces cerevisiae* and human reveals the characteristic anticodon-binding domain linked by a hinge region to the C-terminal catalytic domain, with conserved motifs 1 – 3 [35] (Figs 4A and S4). The *Plasmodium* sequences have a species-specific insertion in the anticodon-binding domain [36], while the *Plasmodium* and yeast sequences have a variable length N-terminal extension with a lysine-rich motif that is thought to bind RNA [36,37]. Previous studies indicated that translation of *Pf*AspRS is initiated from an internal methionine, Met 49 [36].

We used an *E. coli* expression system to generate recombinant native length *Pf*AspRS (49-626) (PF3D7_0102900) and the equivalent predicted native length *Pv*AspRS (51-631) (PVX_081610). Full-length *Hs*AspRS (1–501) (NP_001340.2) was also generated to enable a comparison of the susceptibility of the *Plasmodium* and human AspRS enzymes to reaction hijacking. Following removal of the His-tags, gel filtration yielded dimeric proteins. Given a previous report of poor stability of native length *Pf*AspRS [36], we characterized the preparation by mass spectrometry and analytical ultracentrifugation, showing that *Pf*AspRS exists as a homogenous dimer in solution (S5A and S5B Fig).

We assessed the ability of the recombinant aaRSs to consume ATP in the initial phase of the aminoacylation reaction, using the Kinase GLO assay, as previously described [17]. In initial studies, we found that laboratory prepared *E. coli* tRNA (*Ec*tRNA) is effective as a substrate for all three enzyme preparations, which facilitated the comparison. In the absence of tRNA, very little ATP is consumed by each of the enzymes (S5C Fig). Addition of *Ec*tRNA substantively increased the level of ATP consumption (S5C Fig), consistent with productive aminoacylation.

### Recombinant *P. falciparum* and *P. vivax* tRNA synthetases are thermally stabilized upon formation of Asp-DACM and Asp-AMS adducts

Upon incubation in the presence of substrates, *i.e.*, Asp, ATP and *Ec*tRNA, recombinant native length *Pf*AspRS, *Pv*AspRS and *Hs*AspRS exhibited melting temperature ($T_m$) values of 43.0°C, 43.4°C, and 49.1°C, respectively (Fig 4B–G and S1 Table). These $T_m$ values are 2–3°C lower than for the respective apo AspRSs (S1 Table), which may be due to binding of the SYPRO Orange dye by the tRNA preparation, as reported previously [38].

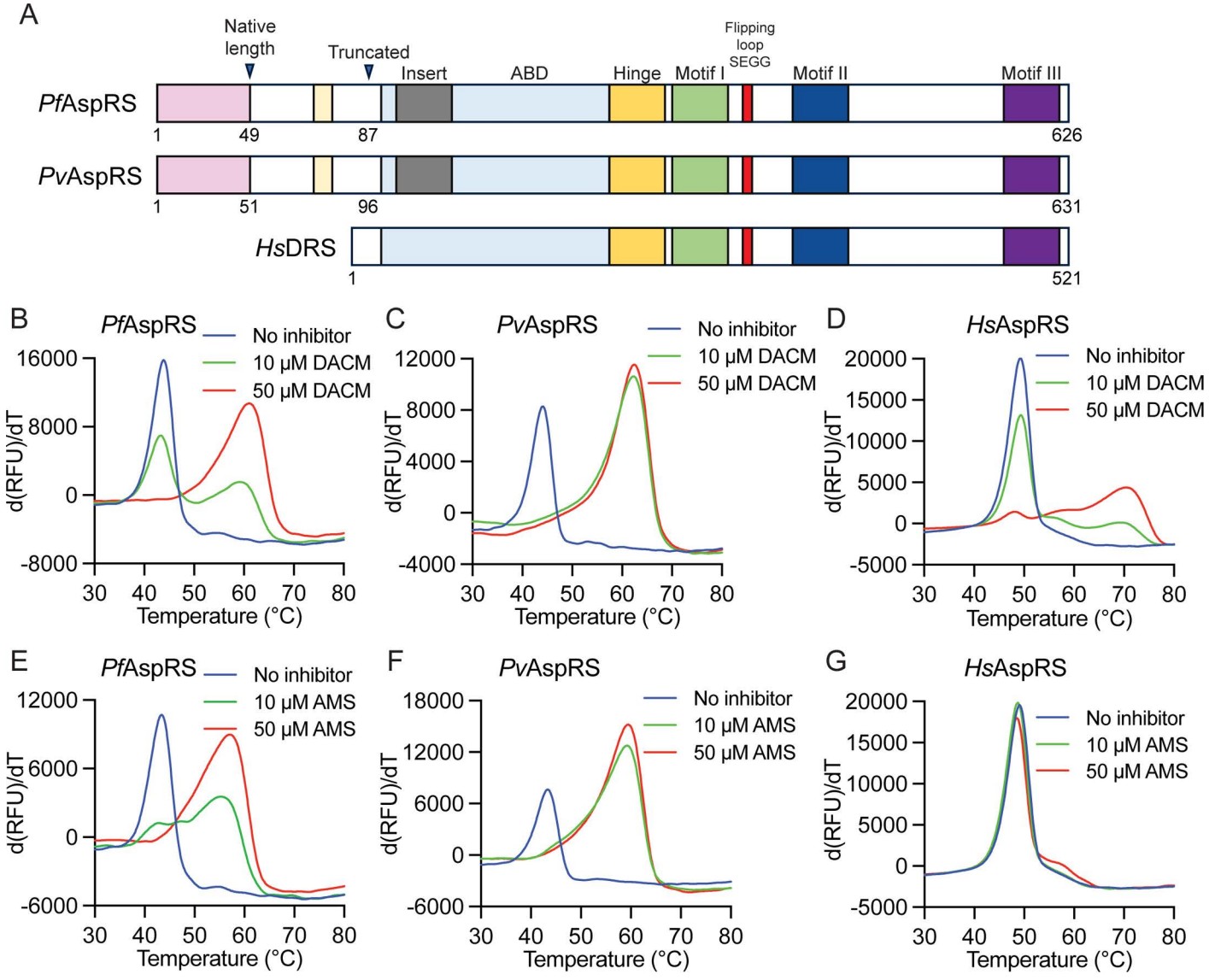

**Fig 4. Thermal stabilization of native length AspRS enzymes by DACM and AMS.** (A) Schematic diagram of AspRS enzymes from *P. falciparum*, *P. vivax* and *Homo sapiens*. Native length *Pf*AspRS and *Pv*AspRS are initiated from internal methionines, M49 and M51, respectively. The anticodon-binding domain (ABD), the hinge region, Motifs I-III, and the flipping loop are indicated. Truncated *Pf*AspRS and *Pv*AspRS constructs comprise residues 87-626 and 96-631, respectively. (B-G) First derivatives of melting curves for native length *Pf*AspRS (B,E), *Pv*AspRS (C,F) and *Hs*AspRS (D,G) (1.5 µM) after incubation at 37°C for 3 h with 10 µM ATP, 20 µM Asp, 80 µM *Ec*tRNA, with 10 or 50 µM DACM (B-D) or 10 or 50 µM AMS (E-G). Data are representative of three independent experiments.

When *Pf*AspRS was incubated with substrates in the presence of 50 µM DACM, the $T_m$ value increased by 18.1°C (to 61.1°C), consistent with the formation of a very tightly bound Asp-DACM adduct (Fig 4B and S1 Table). At a lower DACM concentration (10 µM), two peaks were evident indicating incomplete conversion to the stabilized form of *Pf*AspRS. *Pv*AspRS appears to be even more efficient at forming the Asp-DACM adduct, with the $T_m$ value shift of 19.0°C (to 62.4°C) already largely complete with 10 µM DACM (Fig 4C). By contrast, very little stabilization of *Hs*AspRS was observed at 10 µM DACM, but an emerging population of adduct-bound *Hs*AspRS was observed at 50 µM (Fig 4D). As expected,

incubation with synthetic Asp-DACM increased the $T_m$ values of *Pf*AspRS, *Pv*AspRS and *Hs*AspRS to a similar extent (S5D–F Fig and S1 Table).

Incubation of the recombinant *Pf*AspRS and *Pv*AspRS enzymes with AMS plus substrates also led to marked stabilization, with increases in $T_m$ values to 57.0°C and 59.5°C, respectively. By contrast, *Hs*AspRS was not stabilized by AMS, under the conditions of this experiment (Fig 4E–G and S1 Table). Taken together, these data suggest that *Hs*AspRS is less susceptible to reaction hijacking than *Pf*AspRS and *Pv*AspRS.

## Structure of apo *Pv*AspRS

Attempts to crystallize native length *Pf*AspRS and *Pv*AspRS resulted in poorly diffracting crystals. An AlphaFold analysis predicted that the N-terminal extension is likely to be flexible (S6A Fig), which may impede crystallization. We therefore generated truncated *Pv*AspRS (96–631), lacking the N-terminal extension. *Pv*AspRS (96–631) appears to be capable of catalyzing productive aminoacylation, as indicated by a marked increase in ATP consumption in the presence of *Ec*tRNA (S6B Fig). *Pv*AspRS (96–631) also appears capable of generating the Asp-DACM adduct when incubated with 50 µM DACM in the presence of ATP, Asp and *Ec*tRNA, as indicated by an 18.4°C increase in the $T_m$ value (S6C Fig and S1 Table). Incubation with the synthetic Asp-DACM adduct led to a 20.1°C increase in the $T_m$ value (S6D Fig and S1 Table). These $T_m$ values are similar to those for native length *Pv*AspRS, indicating that the core *Pv*AspRS construct binds the Asp-DACM adduct with similar affinity.

A Morpheus II screen (Molecular Dimensions) was used to generate crystals of *Pv*AspRS (96–631), both as the apo protein and in complex with Asp-DACM, Asp-AMP and Asp-AMS. We solved the structure of apo *Pv*AspRS and refined it to 2.1 Å resolution. Data collection and refinement statistics are summarized in S3 Table. The structure of the apo dimer is presented in Fig 5A, with features highlighted in Chain B, revealing a typical Type II aaRS with an N-terminal β-barrel anticodon-binding domain connected via a hinge to a larger C-terminal catalytic domain that adopts an α-β fold. Motif I is involved in the dimer interface. Motif II and motif III are integral components of the catalytic pocket.

The flipping loop, which plays an important role in substrate binding [39,40], is supported by a β-hairpin structure (Fig 5A, magenta). In apo *Pv*AspRS, residues SEGG (351–354) of the Chain A flipping loop are not resolved (S7A Fig). In Chain B, the flipping loop can be built (Fig 5B). The SEGG loop interacts with a conserved residue Q371, as well as S350, N356 and A357 (S7B Fig). B-factor analysis suggests the SEGG loop is more mobile than neighboring regions of the protein (Fig 5C).

## Structure of Asp-DACM-bound *Pv*AspRS

We solved the structure of *Pv*AspRS with bound synthetic Asp-DACM and refined it to 2.4 Å resolution (S3 Table). Well-defined electron density was observed for the adduct in both chains A and B (Fig 5D). The Asp-DACM is located in the active site pocket of the catalytic domain and interacts with a number of amino acid residues (Fig 5E). The adenine moiety is stacked between H405 and Y409 from motif II. H405 moves from its position in the apo protein to form a parallel π stack with the adenine ring (Figs 5B and S7A). Other motif II loop residues, R396 and E398, also adopt different conformations in the Asp-DACM structure. R396 is highly conserved in prokaryotic and eukaryotic enzymes; and is known to stabilize the transition state during tRNA charging [41]. In both chains, R396 extends to interact with the aspartate of Asp-DACM (Figs 5B,5E and S7A). Similarly, E398 repositions to form an H-bond with the adenine. The aspartic acid moiety of Asp-DACM also interacts with Q374, R561, and K377 (Fig 5E).

As for apo *Pv*AspRS, the flipping loop in Chain B of Asp-DACM-bound *Pv*AspRS is well defined. Residue E352 (of the SEGG motif in Chain B) repositions to interact with the amino group of the aspartate, leading to a small movement towards the ligand (Fig 5B). In Chain A of Asp-DACM-bound *Pv*AspRS, two additional residues of the flipping loop (G353 and G354) are resolved compared with the apo protein (S7C Fig).

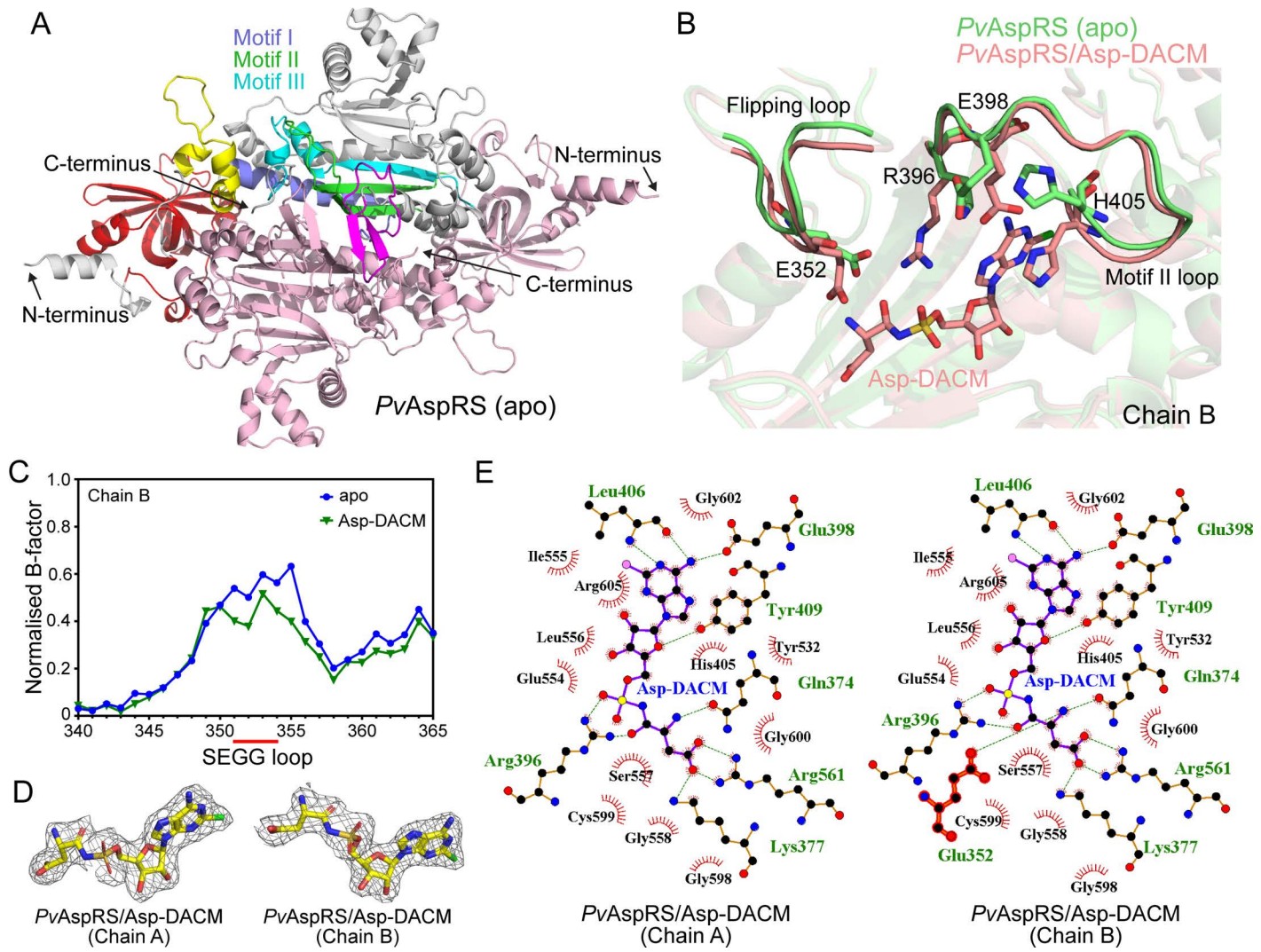

**Fig 5. Comparison of the crystal structures of apo and Asp-DACM-bound *Pv*AspRS reveals binding site interactions.** (A) Ribbon diagram of apo *Pv*AspRS (96-631) with Chain A in salmon and Chain B with domains indicated. The anticodon-binding domain (ABD, red), hinge region (yellow), motif I (violet), motif II (green), motif III (cyan) and flipping loop (magenta) are highlighted, with other regions in grey. (B) Overlay of ribbon representations of the flipping loop and the motif II loop of chain B of apo and Asp-DACM-bound *Pv*AspRS. The sidechain of residue R396 of apo *Pv*AspRS is not fully built due to insufficient density. (C) B-factor analysis of the B chain flipping loops of apo *Pv*AspRS and Asp-DACM-bound structures. The x-axis shows residue number. (D) $2F_o$-$F_c$ maps contoured at 2 σ (mesh surface) showing electron density supporting the position of Asp-DACM bound to chains A and B. (E) Ligplots of Asp-DACM active site interfaces for the A and B chains. Hydrogen bonds and salt bridges are depicted with dashed (green) lines. Other interactions between protein and ligand are indicated by red arcs.

## Structure of Asp-AMS-bound *Pv*AspRS

We also solved the structure of *Pv*AspRS with bound synthetic Asp-AMS and refined it to 1.8 Å resolution, where well-defined electron density was observed for the adduct (S3 Table and Fig 6A). Asp-AMS displays similar interactions to Asp-DACM (Fig 6B), consistent with their similar ability to thermally stabilize *Pv*AspRS. Interestingly, in the Asp-AMS B chain structure, a $Mg^{2+}$ ion coordinates the interaction between the sulfamate oxygen and E554 (Fig 6B). E352 interacts with the amino group of the aspartate in both chains, leading to stabilization of both flipping loops (Fig 6C). B-factor

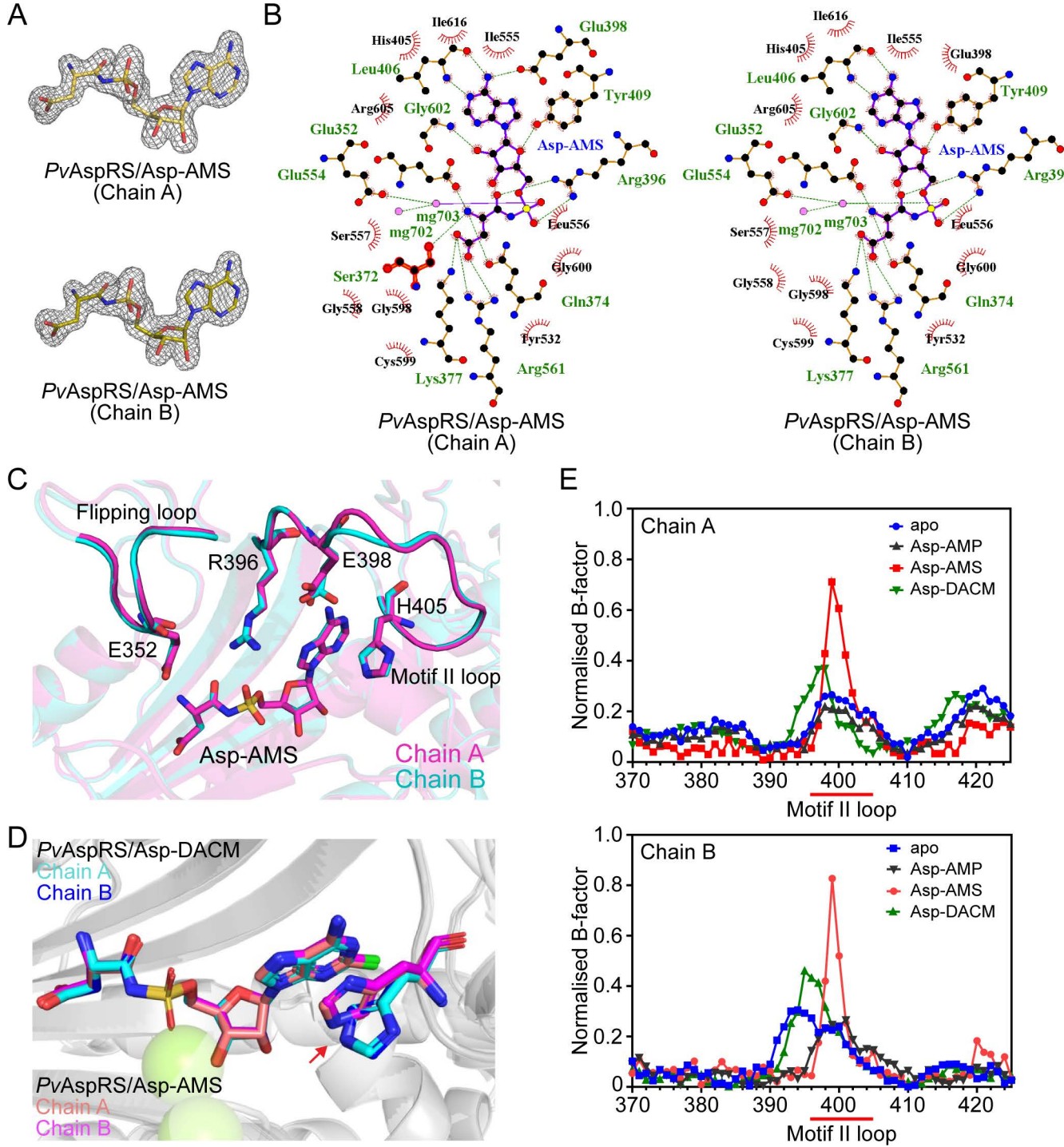

**Fig 6. Crystal structure of Asp-AMS bound *Pv*AspRS.** (A) $2F_o$-$F_c$ maps contoured at 2 σ (mesh surface) showing electron density supporting the position of Asp-AMS bound to chain A and chain B. (B) Ligplots of Asp-AMS/active site interfaces for the A and B chains. Hydrogen bonds and salt bridges are depicted with dashed (green) lines. Other interactions between protein and ligand are indicated by red arcs. (C) Overlay of chains A and B of Asp-AMS-bound *Pv*AspRS showing the flipping loop and the motif II loop. (D) Overlay of chains A and B of Asp-AMS-bound and Asp-DACM-bound *Pv*AspRS, showing the different positions of H405. (E) B-factor analysis of the motif II loops of the apo and ligand-bound structures. The x-axis shows residue number.

analysis reveals that the B chain flipping loop in Asp-AMS bound *Pv*AspRS exhibits higher stability than in the apo structure (S7C Fig). Interestingly, H405 in the motif II loop adopts a different orientation to the Asp-DACM-bound structure (Fig 6D). It forms a less intimate contact with the adenine ring that may be mediated by CH-π or cation-π interactions. Notably, B-factor analysis indicates that, in both chains, the motif II loop in Asp-AMS-bound *Pv*AspRS exhibits greater mobility than other structures (Fig 6E). Thus, the interactions of Asp-DACM and Asp-AMS with the active site are subtly different.

### Structure of Asp-AMP-bound *Pv*AspRS

We also solved the structure of *Pv*AspRS with the natural intermediate, Asp-AMP, and refined it to 2.1 Å resolution (S3 Table and S8A Fig). Asp-AMP is located in the equivalent position within the catalytic domain as Asp-DACM and Asp-AMS; and many of the interactions are similar (S8B Fig). Of interest, in the Asp-AMP structure, H405 occupies the same position as in the apo protein and does not form a π-π interaction with the ligand (S8C Fig). Similarly, E398 from the motif II loop remains in a similar position as in the apo protein, directed away from the active site (S8C Fig, right panel). This orientation is not consistent with E398 interacting with the amino group in the adenine ring, as is the case for the Asp-DACM and Asp-AMS structures (compare Figs 6C and S8C, right panel). As for apo *Pv*AspRS, the flipping loop SEGG residues are resolved in Chain B, but not in Chain A (S8C Fig). B-factor analysis shows that the B chain flipping loop in the Asp-AMP-bound structure has similar flexibility to the apo and Asp-DACM structures (S7C Fig); but is less ordered than the Asp-AMS structure (S7C Fig).

### Comparison of apo *Pv*AspRS with apo *Hs*AspRS

The structure of apo *Hs*AspRS (PDB: 4j15) has been published previously [42]. The two chains of the dimer are equivalent in the human structure. Superimposition of apo *Hs*AspRS with apo *Pv*AspRS (96–631) chain B reveals conservation of the overall structure, particularly in the catalytic and anticodon-binding domains (Fig 7A). By contrast, some regions of differential flexibility are observed (Fig 7B and 7C). Of particular interest, residues I224-Q248, comprising the flipping loop and flanking residues, are disordered in *Hs*AspRS. By contrast, in the equivalent region of apo *Pv*AspRS (L347-Q371), the flipping loop and flanking β-hairpin are both well-defined in Chain B (Fig 7B), while in apo Chain A, only the flipping loop itself is not resolved. Similarly, a loop (R273 to H282) in the middle of motif II, including residues that interact with the end of acceptor stem on the major groove [42–44], is not resolved in apo *Hs*AspRS, but the equivalent region (R396-H405) is ordered in apo *Pv*AspRS (Fig 7B). *Pv*AspRS exhibits an insert in the anticodon-binding domain that is not present in *Hs*AspRS. The insert is disordered and not resolved in the *Pv*AspRS structure (Fig 7C).

### Comparison of apo and Asp-AMP-bound structures of *Pv*AspRS and *Ec*AspRS

We also compared the apo and Asp-AMP-bound structures of *Pv*AspRS (96–631) with the equivalent *E. coli* AspRS structures. In apo *Ec*AspRS (PDB: 1eqr; [45]), the flipping loop is resolved, but with a high B factor. The apo *Ec*AspRS flipping loop (molecule 1/chain A) adopts an "open" conformation that is displaced compared with the flipping loop in chain B of *Pv*AspRS (Fig 7D, top panel). By contrast, in the Asp-AMP-bound *Ec*AspRS structure (PDB: 1il2; [39]), the flipping loop adopts a "closed" structure (Fig 7D, bottom panel) that is similar to that of Asp-AMP-bound *Pv*AspRS. In apo *Pv*AspRS, the SEGG loop is stabilized by interactions with the conserved residues, Q371 and S372, as well as residues Y532, S350 and A357 (Fig 7E). These latter residues are not conserved in *Ec*AspRS, and the equivalent PEGA loop only interacts with Q192, S193 and two water molecules. Thus, differences in the sequences of *Pv*AspRS and *Ec*AspRS in this region likely underpin the observed structural differences, enabling the apo *Pv*AspRS chain B flipping loop to adopt a conformation that is poised to form a "closed" structure.

## Discussion

Natural product nucleoside sulfamate antibiotics were identified nearly 70 years ago. Nucleocidin, a fluorinated nucleoside, was isolated from a soil dwelling *Streptomyces* [46,47]. At the time, it was notable as the first compound, natural

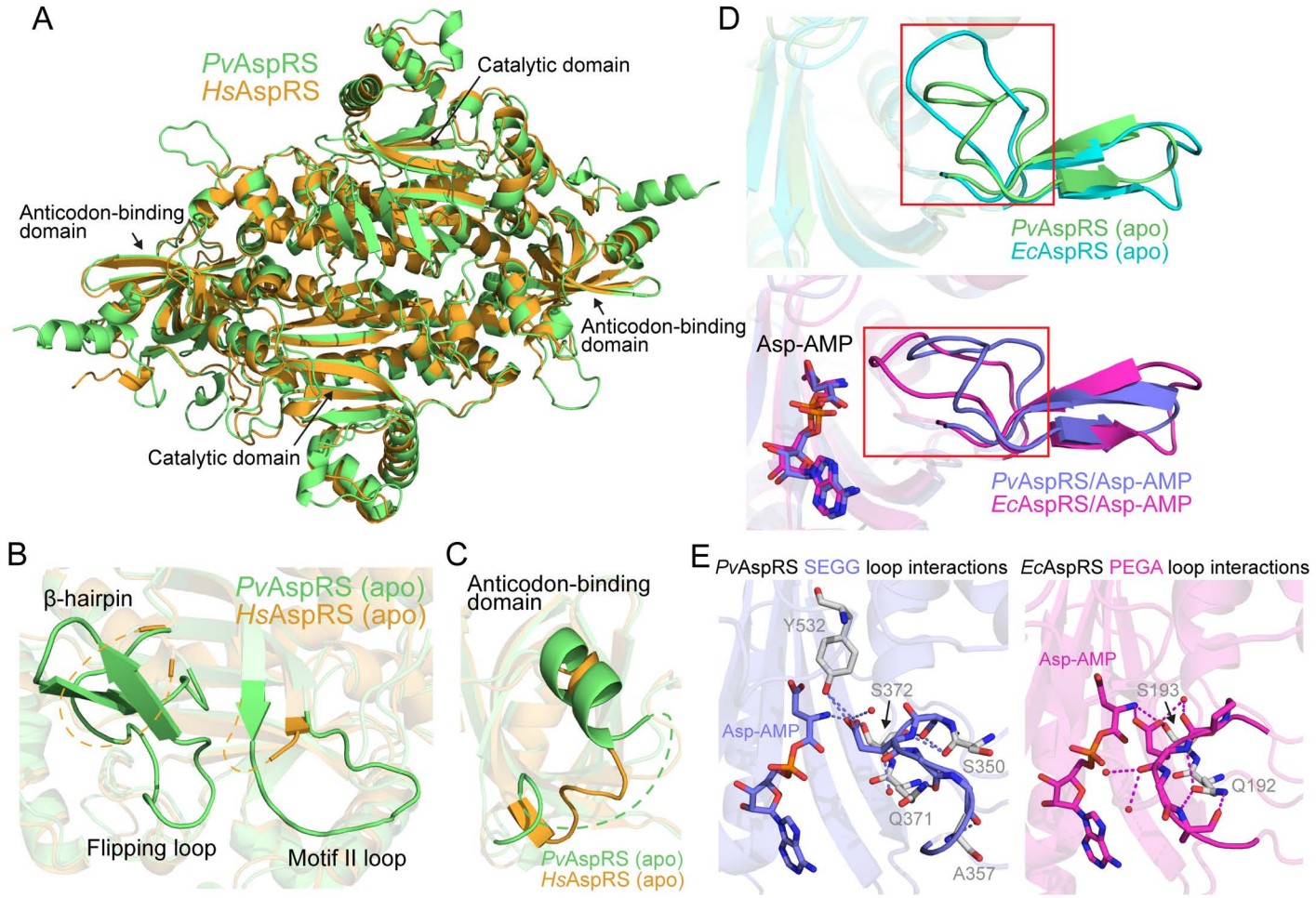

**Fig 7. Comparison of the crystal structures of apo *Pv*AspRS with *Hs*AspRS and *Ec*AspRS.** (A) Overlay of apo *Pv*AspRS and apo *Hs*AspRS (PDB: 4j15) illustrating the conservation of the overall structure. (B) The flipping loop and associated β-hairpin region, and the motif II loop, are unstructured in apo *Hs*AspRS and more ordered in apo *Pv*AspRS (Chain B). (C) Overlay of the anticodon-binding domains of *Pv*AspRS and *Hs*AspRS. The *Plasmodium*-specific insertion is not resolved. (D) Overlay of the apo (top panel) and Asp-AMP-bound (bottom panel) *Pv*AspRS (chain B) and *Ec*AspRS (PDB: 1eqr chain A and 1il2 chain B) active sites illustrating the different positions of the flipping loops. (E) Comparison of the flipping loop interactions of Asp-AMP-bound *Pv*AspRS (SEGG; chain B) and *Ec*AspRS (PEGA; PDB: 1il2, chain B). SEGG interactions are depicted on the left panel. PEGA interactions are depicted in the right panel. SEGG and PEGA loop residues are shown as sticks and the interacting residues are shown in grey.

or synthetic, with an N-unsubstituted sulfamate ester group. Later, a chlorine-containing nucleoside sulfamate, DACM, was isolated from *Streptomyces* [19,48]. These AMP mimics were shown to have activity against Gram-positive and Gram-negative bacteria and trypanosomes [18,19,46]. The synthetic AMP mimic, AMS, was also shown to be active against bacteria [22]. The mechanism of action of the natural nucleoside sulfamates was not clear, but they were reported to inhibit protein synthesis [20–22,49].

Here we show that DACM and AMS cause potent inhibition of the growth of *P. falciparum*, potentially due to inhibition of multiple aaRSs. These compounds also show inhibitory activity against a mammalian cell line. This is in agreement with a previous study showing that DACM causes toxicity in mice [48]. Such toxicity will limit the development of these broad-specificity nucleoside sulfamates as antimalarials. Nonetheless, AMS and DACM represent useful starting points for the synthesis of more selective compounds that can target malaria parasite AspRS.

Treatment of *P. falciparum* cultures with DACM or AMS inhibits protein translation and induces phosphorylation of eIF2α, as expected upon inhibition of aaRSs and the consequent accumulation of uncharged tRNA. eIK1, the *P. falciparum* homologue of GCN2, is the responsible kinase [50,51]. Accordingly, treatment of eIK1 knockout parasites with DACM or AMS, did not result in eIF2α phosphorylation. Taken together, these data indicate that AMS and DACM exert their activity by inhibiting aaRSs.

Reaction hijacking compounds induce the target aaRS to synthesize amino acid adducts with a stable sulfamoyl bond. The adduct binds tightly in the active site and replaces the natural amino acid adenylate, which has a labile aminoacyl bond. In previous studies, we showed that ML901/ ML471 and OSM-S-106, specifically target *Pf*TyrRS (class I) and *Pf*AsnRS (class II), respectively [15–17]. By contrast, AMS targets several aaRSs, with *Pf*TyrRS being the most susceptible enzyme [15].

Based on the adduct signals identified by mass spectrometry, *Pf*AsnRS appears most susceptible to DACM. *Pf*AspRS, *Pf*ThrRS, *Pf*SerRS and *Pf*LysRS also appear to be susceptible enzymes while the adduct levels generated by *Pf*HisRS and *Pf*PheRS are lower. We chose *Pf*AspRS for additional studies, as this enzyme has not been targeted for inhibitor development previously in *Plasmodium*; and no crystal structures were available before our study. *Plasmodium* AspRS is a Class IIb aaRS that exhibits unusual sequence features compared with human AspRS [36], including initiation from an internal methionine, and an insert in the anticodon-binding domain. *Plasmodium* AspRS also exhibits an extended N-terminal region, with a lysine-rich motif that is thought to facilitate tRNA binding [36]. *Hs*AspRS has a much shorter (21-residue) N-terminal extension that may also be involved in tRNA binding [52,53].

We generated recombinant, native length *Pf*AspRS (49–626) and the equivalent predicted native-length *Pv*AspRS (51–631), as well as a truncated construct of *Pv*AspRS (96–631), lacking the N-terminal extension. We also generated *Hs*AspRS for comparison. Each of these constructs appear to be capable of catalyzing aminoacylation of tRNA, as judged by a substantive increase in ATP consumption upon addition of *Ec*tRNA. These data suggest that the N-terminal extension is not essential for enzyme activity.

Thermal stabilization experiments suggest *Pv*AspRS and *Pf*AspRS catalyze the formation of Asp-DACM and Asp-AMS adducts via the reaction hijacking mechanism. By contrast, *Hs*AspRS appears to be less susceptible to hijacking, particularly by AMS. This is consistent with previous studies showing that *Pf*TyrRS and *Pf*AsnRS are more susceptible to hijacking than their human counterparts [15–17].

We solved the first structures of AspRS from *Plasmodium. Pv*AspRS (96–631) was successfully crystallized as the apo protein, in complex with the natural intermediate, and with synthetic Asp-AMS and Asp-DACM, allowing insights into the binding determinants. As anticipated, *Pv*AspRS adopts a dimeric structure with features typical of Type II aaRSs. The *Plasmodium*-specific insertion in the anticodon-binding domain (residues 137 – 167) is located at the protein surface and resides 151–177 are unresolved. This region is distal to the predicted tRNA anticodon-binding surface and the role of this insertion remains unclear.

AspRSs have been studied extensively in bacteria, yeast and humans. The active site comprises binding pockets for Asp, AMP and the 3′ end of tRNA. In apo *Hs*AspRS and *Pyrococcus kodakaraensis* AspRS, the flipping loop and flanking residues are disordered [40,42]. In the apo forms of the yeast and *E. coli* AspRS, the flipping loop adopts a defined open conformation [45,54]. In each of these cases, following binding of the amino acid substrate, the flipping loop adopts a closed lid-like conformation that contributes to positioning the amino acid in a state ready for attack [39,40]. The closed loop also prevents access of the terminal tRNA adenosine until the adenylate intermediate is formed. The flipping loop then repositions to allow access of the tRNA 3′ end; and it then anchors the A76 base to promote the transfer step [39].

In contrast to many other species studied to date, *Pv*AspRS exhibits structural asymmetry even in the apo state. In *Pv*AspRS chain A, the flipping loop is disordered, while in chain B, the density could be mapped, although it exhibits higher B-factor values, indicating some flexibility of the loop. Interestingly, only minor shifts are observed after binding of ligands suggesting that the apo *Pv*AspRS chain B flipping loop is poised to adopt a closed position. Binding of Asp-AMP, Asp-DACM or Asp-AMS causes modest repositioning E352 (of SEGG), allowing interaction with the amino group of the

aspartate. In chain A of Asp-AMS-bound *Pv*AspRS, the flipping loop becomes structured and E352 engages with the ligand, while in Asp-DACM *Pv*AspRS chain A, only the GG of the flipping loop SEGG residues are resolved.

Upon binding of either Asp-DACM or Asp-AMS, changes are also observed in the motif II loop. E398 extends towards the adenine and forms an interaction with the amino group. In contrast, in Asp-AMP-bound *Pv*AspRS, E398 remains in a similar position as in the apo protein, directed away from the active site. The tighter binding of Asp-DACM or Asp-AMS is also likely enhanced by the stable sulfamoyl bond of the adducts, which prevents cleavage and effectively locks *Pv*AspRS in the adduct-bound state. These differences may contribute to the tighter binding of the adducts. Of interest, H405 adopts different orientations in the Asp-DACM or Asp-AMS structures, forming, respectively, either a parallel π-stack or a T-shaped interaction, that may be CH-π or cation-π in nature. These differences are associated with different flexibility of the motif II loop. The chloro group of DACM is expected to make the adenine ring more electron deficient, which will enhance the π-π interaction between H405 and the nucleobase.

In apo *Hs*AspRS, the motif II loop (R273 to H282) is not resolved. This loop includes residues that interact with the 3' end of the tRNA acceptor stem [43,44,54]. By contrast, in apo *Pv*AspRS, the equivalent region (R396-H405) is ordered. Enhanced binding of aminoacylated tRNA could increase the longevity of the product-bound form of the enzyme, which could predispose *Pv*AspRS to attack by nucleoside sulfamates [15].

In conclusion, we have identified DACM as a new reaction hijacking inhibitor, with potent activity against *P. falciparum*. The *Plasmodium* AspRS is revealed here as a new target for antimalarials that is more susceptible to reaction hijacking than its human counterpart. Differences in the conformation and flexibility of the flipping loop and the motif II loop may underpin the differential susceptibility between parasite and host enzymes. Further work is needed to identify alternative nucleoside-mimicking scaffolds that provide more selective targeting of *Pf*AspRS, as has been achieved for *Pf*TyrRS [15,16] and *Pf*AsnRS [17]. Such selective inhibitors could be developed as future antimalarial compounds.

## Methods

### Ethics statement

The use of human RBCs and serum in this study was ethically approved by the Office of Research Ethics and Integrity (The University of Melbourne). RBCs and serum were acquired from the Australian Red Cross Lifeblood service, Melbourne, Australia. Reference number: 2022-25738-35573-3.

### Inhibition of the growth of *P. falciparum* cultures

Parasite-infected RBCs (3D7 strain [28]) were cultured in RPMI-HEPES containing 0.25% AlbuMAX II and 5% human serum, as described previously [55]. Sorbitol-synchronized ring stage parasites at 1% parasitaemia and 0.2% haematocrit (were incubated with DACM or AMS for 72 h and viability was assessed in the next cycle by flow cytometry, after labelling with 2 μM SYTO 61 (Thermo Fisher Scientific), as previously described [29]. The parasitemia was normalized to untreated and "kill-treated" controls, treated with 10x $IC_{50}$ concentration of each compound, for 72 h. Cells were pelleted at 400 *g* for 1.5 min and were incubated with 2 μM SYTO 61 in phosphate-buffered saline (PBS; Molecular Probes, Life Technologies) at RT for 15 min. Nine volumes of PBS were added to the cells (final SYTO 61 concentration: 0.2 μM) and incubated for a further 30 min at RT. The samples were analyzed by BD Biosciences FACSCanto II flow cytometer using the APC channel where the forward and side scatter was used to gate total cells. Data were processed by BD FACSDiva Software and FlowJo. The half maximal inhibitory concentration ($IC_{50}$) was determined using nonlinear regression (curve fit) in GraphPad Prism.

### Cytotoxicity of PM03 against HepG2 cells

The HepG2 (Human Caucasian hepatocyte carcinoma) cell line was procured from Cell Repository NCCS, India, and cultured in Dulbecco's Modified Eagle Medium (DMEM) supplemented with 10% Fetal Bovine Serum (FBS), 4 mM glutamine

and 50 µg/mL penicillin-streptomycin, in a humidified incubator at 37°C with 5% $CO_2$. Using an assay procedure modified from [56], 10,000 cells per well were incubated in 96-well plates for 24 h at 37°C with 5% $CO_2$, to allow attachment. The medium was removed; and cells were treated with fresh medium (100 µL) containing either vehicle or serial dilutions (2-fold) of the compound, prepared in DMEM, with 2% FBS, in triplicate. Cells treated with 20% DMSO were used as a kill control. The cells were incubated for 48 h at 37°C with 5% $CO_2$. The growth medium was aspirated, and 100 µL of 0.5 mg/mL MTT solution was added to each well. After 3 h at 37°C, with 5% $CO_2$, the MTT solution was removed and 100% DMSO was added to dissolve formazan crystals, with shaking, in the dark, at 37°C for 15 min. The absorbance was measured at 570 nm using a SpectraMax M3 microplate reader. The absorbance at 630 nm was subtracted to correct for background noise. Graph Prism 9 was used to generate the dose-response curve using non-linear regression analysis (variable slope).

### Protein translation assay

*P. falciparum* Cam3.II[Rev] [57] trophozoite (30–35 h p.i.) infected RBCs (0.2% hematocrit and 1% parasitemia) were exposed to the relevant compounds for 6 h. The cells were labelled with O-propargyl-puromycin (OPP (Abcam); 4 µM) in the final 2 h, washed two times (PBS + 3% human serum) and then fixed and permeabilized, as described previously [17]. The click reaction was performed for 1 h at 37°C in the presence of $CuSO_4$ (0.1 mM), tris-hydroxypropyltriazolylmethylamine (THPTA; 0.5 mM) and sodium ascorbate (5 mM) to bring about azide-alkyne cycloaddition to Alexa Fluor 488 azide (0.1 µM). Pellets were washed four times and resuspended in PBS + 3% human serum containing 25 µg/mL propidium iodide (Invitrogen). Flow cytometry (FACS Canto II; BD Biosciences, San Jose, CA) was performed using a BD FACSDiva (version 8.0) and FlowJo (version 10.9). Side scatter height (SSC-H) and forward scatter area (FSC-A) density plots were used to gate the total cell population. FSC-A and forward scatter width (FSC-W) plots were used to gate the single cell population. The FITC and PerCP-Cy5.5-H channels were used to detect the Alexa Fluor 488 and propidium iodide (PI) positive populations, representing parasitized RBCs. The same 6-h drug treatment conditions were set up in parallel for viability assessment. Cells were washed intensively 3 x with Complete Culture Medium (CCM) to remove inhibitors and returned to culture. Parasite viability was assessed in the next cycle as previously described above.

### Phospho-eIF2α analysis

Trophozoite-infected RBCs (26 – 32 h p.i.; 2.5% hematocrit, 5–6% parasitemia) of the Cam3.II[Rev] line [57] or an *eIK1* knockout line [30] (kindly provided by Prof Christian Doerig, RMIT University), were incubated with 0.2 µM borrelidin, 1 µM ML901, 1 µM AMS, 1 µM DACM, or DMSO, for 3 h. The RBC pellet was washed 3 x in PBS + cOmplete EDTA-free Protease Inhibitor Cocktail. The pelleted cells were lysed by resuspension in PBS + 0.05% saponin, on ice. Washed pellets were resuspended in Bolt LDS sample buffer plus reducing agent and subjected to Western analysis as described previously [17]. Primary antibodies: rabbit anti-phospho-eIF2α (Cell Signaling Technology-119A11; 1:1,000); polyclonal mouse anti-*Pf*BiP, generated using recombinant *Pf*BiP at the WEHI Antibody Services (1:1,000). Secondary antibodies: goat anti-rabbit IgG-peroxidase (Chemicon-A132P; 1:20,000), goat anti-mouse IgG-peroxidase (Chemicon-A181P; 1:50,000). Membranes were washed three times in 1xPBS + 0.1% Tween 20. Chemiluminescence was detected using the Bio-Rad ChemiDoc MP imaging system.

### Sample preparation to identify amino acid DACM conjugates

Late trophozoite stage *P. falciparum* (3D7 strain) culture samples was exposed to 10 µM DACM for 3 h. The parasite-infected RBCs were lysed with 0.1% saponin in PBS and the parasite pellet was washed 3 times with ice-cold PBS. Cell pellets were kept on ice and resuspended in water as one volume, followed by the addition of five volumes of cold chloroform-methanol (2:1 [vol/vol]) solution. Samples were incubated on ice for 5 min, subjected to vortex mixing for 1 min and centrifuged at 13,500 x *g* for 10 min at 4°C to form 2 phases. The top aqueous layer was transferred to a new tube

and subjected to LC-MS analysis. The synthetic Asp-DACM standard was processed in the same way. Data analysis was performed using Xcalibur (version 4.4).

## High-performance liquid chromatography (HPLC) and mass spectrometric (MS) analyses

Samples were analyzed by reversed-phase ultra-high performance liquid chromatography (UHPLC) coupled to tandem mass spectrometry (MS/MS) employing a Vanquish UHPLC linked to an Orbitrap Fusion Lumos mass spectrometer (Thermo Fisher Scientific, San Jose, CA, USA) operated in positive ion mode, modified from a previous procedure [15]. Solvent A was 0.1% formic acid acetate in water and solvent B was 0.1% formic acid in acetonitrile. 5 µL of each sample was injected onto an RRHD Eclipse Plus C18 column (2.1 × 100 mm, 1.8 µm; Agilent Technologies, USA) held at 50 °C with a solvent flow rate of 350 µL/min. The solvent gradient was as follows [Time (min), B %]: [0, 0], [6, 0], [13,25], [13.1, 99], [14, 99], [14.1, 0], [18, 0]. Mass Spectrometry experiments were performed using a Heated Electrospray Ionization (HESI) source. The spray voltage, flow rate of sheath, auxiliary and sweep gases were 3.5 kV, 25, 5, and 0 'arbitrary' unit(s), respectively. The ion transfer tube and vaporizer temperatures were maintained at 300°C and 150°C, respectively, and the S-Lens RF level was set at 30%. A full-scan MS spectrum and targeted MS/MS for the proton adduct of Asp-DACM or the 20 possible common amino acid-containing inhibitor adducts were acquired in cycles throughout the run. The full-scan MS spectra were acquired in the Orbitrap at a mass resolving power of 120,000 (at *m/z* 200) across an *m/z* range of 200–1500 using quadrupole isolation and the targeted MS/MS were acquired using higher-energy collisional dissociation (HCD)-MS/MS in the Orbitrap at a mass resolving power of 7500 (at *m/z* 200), a stepped normalized collision energy (NCE) of 15, 30 and 45% and an *m/z* isolation window of 1.2.

## Generation of recombinant aaRSs

The gene sequences encoding native length *Pf*AspRS (49–626) (PlasmoDB ID: PF3D7_0102900), native length *Pv*AspRS (51–631) (PVX_081610) and full-length *Hs*AspRS (1–501) (NP_001340.2) were codon-optimized for expression in *Escherichia coli*, synthesized by GenScript, and cloned into pET-11a vector with a histidine tag and TEV cleavage site. Truncated *Pf*AspRS (87–626) and *Pv*AspRS (96–631) genes were amplified from the synthesized plasmids using PCR and cloned into pET-11a and pETM-41 vectors, respectively. The proteins were overexpressed in *E. coli* BL21 (DE3) using 0.05 mM or 0.1 mM IPTG induction at 16°C, overnight. The lysis buffer was 50 mM Tris pH 7.4 (or pH 8.0), 350 mM NaCl, 40 mM imidazole, 0.5 mM tris(2-carboxyethyl)phosphine (TCEP), 1 mg/mL lysozyme, and 1x complete protease inhibitor. Clarified lysate was loaded onto a HisTrap (HP) 5 mL nickel column (Cytiva). Proteins were eluted with a gradient of Buffer B (50 mM Tris pH 7.4, 350 mM NaCl, 500 mM imidazole, 0.5 mM TCEP). *Pf*AspRS was eluted at ~40% Buffer B. The eluted sample was dialyzed against 50 mM Tris-HCl pH 7.4, 150 mM NaCl, 25 mM imidazole (or 200 mM imidazole), and 1 mM TCEP, with TEV protease to remove the N- terminal histidine tag. The sample was concentrated and subjected to size exclusion chromatography using buffer containing 25 mM or 50 mM Tris-HCl pH 8.0, 150 mM NaCl, and 0.5 or 1 mM TCEP.

## Characterization of native length *Pf*AspRS

For analytical ultracentrifugation, recombinant *Pf*AspRS was prepared at 1, 0.6, and 0.2 mg/mL in buffer containing 50 mM Tris pH 8.0, 150 mM NaCl, 1 mM TCEP. Samples and buffer (reference solution) were centrifuged at 200,000 g at 20°C, in a Beckman Coulter XL-I analytical ultracentrifuge, equipped with UV-visible scanning optics. Radial absorbance data were monitored and collected at a wavelength of 290 nm. Sedimentation data were fitted to a continuous sedimentation coefficient (c(s)) model, with frictional ratios estimated using SEDFIT software [58].

For mass spectrometry, purified protein (5 µg) in 25 mM Tris pH 8.0, 150 mM NaCl, 1 mM TCEP was subjected to an Agilent 1200 HPLC, equipped with a C18 column, connected to an Agilent 6220 Accurate-Mass electrospray ionization time-of-flight (ESI-TOF) mass spectrometer. Data acquisition and analysis were performed using Mass Hunter Software (Agilent).

## Preparation of *E. coli* tRNA

Total tRNA from *E. coli* was isolated with modifications from a previous report [59]. *E. coli* BL21(DE3) cells were cultured in 2×Yeast Extract Tryptone medium at 37°C overnight. Cells were harvested by centrifugation, resuspended in diethyl pyrocarbonate (DEPC)-treated water, and lysed by adding TRIzol reagent (Invitrogen) at a 3:1 ratio to the cell suspension, followed by vigorous vortexing. After centrifugation, the aqueous phase was extracted, and acid-phenol: chloroform pH 4.5 (with indole-3-acetic acid (IAA), 25:24:21; Invitrogen) was added. Following another centrifugation step, the supernatant was collected. This step was repeated until a clear interface was observed. Next, LiCl was added to a final concentration of 1 M. tRNA was precipitated with ice-cold isopropanol, then dissolved in DEPC-treated water for further use.

## ATP consumption assay

The consumption of ATP by native length *Pf*AspRS, native length *Pv*AspRS, truncated *Pv*AspRS and full-length *Hs*AspRS was determined using a luciferase-based assay as per the manufacturer's instructions (Kinase-Glo Luminescent Kinase Assay, Promega). Reactions were conducted in 25 mM Tris-HCl (pH 8), 150 mM NaCl, 5 mM $MgCl_2$, 0.1 mg/mL BSA, 1 mM TCEP, with 200 µM L-aspartate, 10 µM ATP, 1 unit/mL inorganic pyrophosphatase and 80 or 160 µM *E.coli* tRNA (if present). Enzyme concentration and incubation time for each experiment are described in the relevant figure legend. Reactions were incubated at 37°C, followed by addition of the Kinase-Glo reagent. Luminescence output was measured using a plate reader (CLARIOstar, BMG LABTECH) and the highest signal within 20 min after addition of reagents was recorded using MARS data analysis software (version 3.32). Assay conditions were optimized to ensure ATP consumption is in the linear range with respect to AspRS concentration. The concentration of ATP was quantified by linear regression using an ATP standard curve (Microsoft Excel).

## Differential scanning fluorimetry (DSF)

The effect of DACM, AMS and Asp-DACM on the thermal stability of AspRS enzymes was assayed as previously described [15]. Briefly, the relevant AspRS (1.5 µM) was incubated in the presence or absence of 10–50 µM DACM or AMS, or 2, 5 or 10 µM Asp-DACM, with 10 µM ATP, 20 µM L-aspartate, 80 µM *Ec*tRNA, in 25 mM Tris-HCl (pH 8), 150 mM NaCl, 5 mM $MgCl_2$, 1 mM TCEP, at 37°C for 3 h. SYPRO Orange (Sigma-Aldrich; 5,000X concentrate in DMSO) was added to the reaction mixture at a final concentration of 5X. 25 µL of the sample was added into each well of a 96-well qPCR plate (Applied Biosystems). The plate was sealed and analyzed using StepOnePlus Real-Time PCR system (Applied Biosystems). The samples were heated from 20°C to 90°C with a 1°C per min continuous gradient. The thermal unfolding curve was plotted as the first derivative curve of the raw fluorescence values. The melting temperature ($T_m$), defined as the peak of the first derivative curve, was used to assess the thermal stability of protein-ligand complexes.

## Crystallization and X-ray diffraction data collection

For crystallization, *Pv*AspRS (96–631) in the apo form, prepared in 20 mM HEPES pH 8.0, 200 mM NaCl, 10% glycerol, and 5 mM beta-mercaptoethanol, was concentrated to 20 mg/mL and incubated without or with bound natural substrates (ATP and L-Asp) or synthetic Asp-AMS or Asp-DACM, with 5 mM $MgCl_2$, at a molar ratio of 1:4–1:30 (*Pv*AspRS monomer/ ligand). Crystallization experiments were performed using the sitting-drop or hanging-drop vapour-diffusion method at 293 K. Initial crystallization screening was carried out in a 96-well plate (Corning, Lowell, Massachusetts, USA) with ViewDrop II seals (SPT LabTech, Melbourn, England) using the commercially available crystallization sparse-matrix screens (SG1, ProPlex, and Morpheus I and II; Molecular Dimensions, UK) [60]. Three different drop ratios were aliquoted using a Mosquito nanolitre dispenser system (TTP LabTech, Melbourn, England) or an NT8 drop setter (Formulatrix) by mixing protein and reservoir solutions at 1:1, 2:1 and 1:2 drop ratios; final volume 150 nL). The crystallization droplets were equilibrated against a 75 µL reservoir solution. Initial crystals of *Pv*AspRS were obtained in the following conditions.

Apo *Pv*AspRS: SG1-D10 (0.2 M lithium sulfate, 0.1 M Bis-Tris pH 6.5 and 25% w/v PEG 3350); *Pv*AspRS (Asp-AMP): SG1-C12 (0.2 M sodium acetate trihydrate, 0.1 M Bis-Tris pH 5.5 and 25% w/v PEG 3350); *Pv*AspRS (Asp-AMS): Morpheus I-A9 (0.06 M divalents (0.3 M magnesium chloride hexahydrate; 0.3 M calcium chloride dihydrate), 0.1 M Buffer System (Tris base and BICINE), pH 8.5 and 30% v/v Precipitant Mix (40% v/v PEG 500 MME and 20% w/v PEG 20000); *Pv*AspRS (Asp-DACM): Morpheus II-B5 (15% (w/v) PEG 3000, 20% (v/v) 1,2,4-butanetriol, 1% (w/v) nondetergent sulfobetaine (NDSB) 256, 0.5 mM manganese chloride, 0.5 mM cobalt chloride, and 0.5 mM zinc chloride). The crystals were cryoprotected using 10–20% glycerol before being flash-cooled in liquid nitrogen. X-ray diffraction experiments were conducted on the MX2 beamline at the Australian Synchrotron [61] or the I03 beamline at the Diamond Light Source, UK (S3 Table).

### Structure determination

Several datasets were collected. Data were indexed and integrated using the XDS software package [62] and scaled using AIMLESS [63]. Alternatively, data were processed using the xia2/DIALS [64] and autoPROC [65] pipeline. Human AspRS (*Hs*AspRS, PDB ID: 4J15 [42]) was used as the phasing model. The initial phases were determined by molecular replacement using *PHASER* [66] or Auto-Rickshaw [67]. The model was further refined using *phenix.refine* from *PHENIX* [68,69] and manually built using *COOT* [70]. Ligands, ions and water molecules were added to their electron densities after several rounds of manual model building and refinement. Structure refinement was performed using non-crystallographic torsion restraints and translation/libration/screw (TLS) refinement with each chain comprising a single TLS group. Restraints for Asp-AMP, Asp-AMS, Asp-DACM were generated using phenix.elbow [71] or GRADE Web Server (Global Phasing Ltd, https://grade.globalphasing.org). Difference density peaks observed near the 2-Cl moiety of Asp-DACM suggested radiation damage in this location during data collection. *MolProbity* [72,73] in *PHENIX* suite was used to evaluate model quality and figures were generated using UCSF Chimera [74] and PyMOL (http://www.pymol.org), including the embedded PyMOL secondary structure assignment. Magnesium ions were identified using CheckMyMetal [75]. Complete data collection and refinement statistics are summarized in S3 Table. LigPlot+ (version 2.2.8) was employed to analyze ligand-protein interactions and to generate 2D graphical maps [76].

### Isotropic B-factor analysis

Isotropic B-factors for the alpha carbons for each residue were extracted from the PDB files in PyMOL Version 2.5.4 [77]. The B-factors were corrected by dividing by the Wilson B-factor of each structure ($B_{Corrected} = \frac{B_{CA}}{Wilson\ B}$) and then normalised using the following equation: $B_{norm} = \frac{B_{corrected} - B_{MIN}}{B_{MAX} - B_{MIN}}$. The resulting normalized, dimensionless B-factor derived values ($B_{norm}$) ranged from 0-1, with higher values indicating greater atomic mobility.

### Chemistry

Synthetic procedures and compound characterizations are provided in S1 Text.

### Supporting information

**S1 Fig. Activity of AMS against *P. falciparum* and HepG2 cells.** (A) Structure of AMS. (B) Sorbitol-synchronized ring stage parasites were subjected to a 72-h drug sensitivity assay with AMS (black circles). Data represent five independent experiments, each performed in duplicate. Cytotoxicity of AMS (white circles) against the HepG2 mammalian cell line in a 48-h exposure assay. Data represent five independent experiments, each performed in triplicate. Error bars indicate SEM. (TIF)

**S2 Fig. Identification of amino acid-DACM- conjugates in *P. falciparum*.** *P. falciparum* cultures were exposed to 10 μM DACM for 3 h. Parasite extracts were subjected to LC-MS/MS to search for DACM-amino acid conjugates. (A-F)

Detected (top panels) and predicted mass spectra of (A) DACM-Asn ($m/z = 495.0808$), (B) DACM-Lys ($m/z = 509.1328$); (C) DACM-Thr ($m/z = 482.0855$); (D) DACM-Ser ($m/z = 468.0699$), (E) DACM-His ($m/z = 518.0968$), (F) DACM-Phe ($m/z = 528.1063$). (G-L). MS/MS spectra of the fragmented ions, including a $m/z$ of 170.0228 found as a fragmented ion of each DACM-amino acid conjugate.
(TIF)

**S3 Fig. DSF and docking analysis for *Pf*TyrRS.** (A) First derivatives of melting curves of *Pf*TyrRS (2.3 µM) after incubation at 37°C for 2 h with 10 µM ATP, 20 µM Tyr, 4 µM *Pf*tRNA$^{Tyr}$, with 5 or 10 µM of AMS or DACM. Data are representative of three independent experiments. (B) A chlorine atom was added to the 2-position of the adenine ring system of Tyr-AMP at the active site of *Pf*TyrRS (PDB: 7ROR) using ChimeraX [78]. Steric overlaps between the chlorine atom and the protein binding pocket are shown as red dashed rods.
(TIF)

**S4 Fig. Sequence alignment of AspRS sequences from different species.** Alignment of AspRS sequences from *P. falciparum* (*Pf*), *P. vivax* (*Pv*), *Homo sapiens* (*Hs*), *Saccharomyces cerevisiae* (*Sc*), reveals a high level of conservation of the three Type II aaRS motifs (I-III, blue, green, purple text), which are involved in ATP binding and dimerization. The hinge region is highlighted in yellow. The *Plasmodium* sequences exhibit a large N-terminal extension with native initiation from an internal methionine (aqua text). The anticodon-binding domain (salmon) has a *Plasmodium*-specific insert (underlined). The flipping loop residues, SEGG, that have previously been shown to undergo dynamic motions that facilitate tRNA binding [40], are in red text. Two loops that are ordered in *Pv*AspRS but disordered in *Hs*AspRS are boxed, namely the flipping loop and flanking β-hairpin structure and the motif II loop.
(TIF)

**S5 Fig. Characterization of native length *Pf*AspRS and stabilization by the Asp-DACM adduct.** (A) Deconvoluted mass spectrum obtained using Agilent Mass Hunter software. The mass of the highest peak (67831 Da) correlates well with the theoretical mass of native length *Pf*AspRS. (B) Sedimentation velocity analysis. Purified native length *Pf*AspRS was diluted to 1, 0.6, and 0.2 mg/mL in buffer containing 50 mM Tris-HCl (pH 8), 150 mM NaCl, and 1 mM TCEP. Samples were subjected to analytical ultracentrifugation. Samples were centrifuged at 200,000 $g$ and monitored at a wavelength of 290 nm. The continuous sedimentation coefficient c(s) was plotted as a function of the sedimentation coefficient (S). (C) ATP consumption by *Pf*AspRS, *Pv*AspRS and *Hs*AspRS in the presence and absence of the *Ec*tRNA. Reagent concentrations: 50 nM *Pf*AspRS and *Pv*AspRS or 100 nM *Hs*AspRS with 10 µM ATP, 200 µM Asp, 1 U/mL pyrophosphatase, 80 µM *Ec*tRNA for *Pf*AspRS and *Pv*AspRS and 160 µM *Ec*tRNA for *Hs*AspRS. Data represent mean ± SEM from four independent experiments. (D-F) Thermal stabilization of native length AspRS enzymes by Asp-DACM. First derivatives of melting curves for native length *Pf*AspRS (D), *Pv*AspRS (E) or *Hs*AspRS (F) (1.5 µM) after incubation at 37°C for 3 h with 2 or 5 µM Asp-DACM. Data are representative of three independent experiments.
(TIF)

**S6 Fig. Structure prediction for full-length *Pf*AspRS and *Pv*AspRS and biochemical analysis of truncated *Pv*AspRS.** (A) AlphaFold predicted structures of full-length *Pf*AspRS (PlasmoDB ID: PF3D7_0102900) and full-length *Pv*AspRS (PlasmoDB ID: PVX_081610). Model confidence is predicted and colored. Blue: Very high (pLDDT > 90), sky blue (90 > pLDDT > 70), yellow (70 > pLDDT > 50), and orange: very low (pLDDT < 50) per-residue model confidence score (pLDDT). (B) ATP consumption by native length *Pv*AspRS (51–631) and truncated *Pv*AspRS (96–631) in the presence and absence of the *Ec*tRNA. 50 nM *Pv*AspRS was incubated with 10 µM ATP, 200 µM Asp, 1 U/mL pyrophosphatase, ±80 µM EctRNA in 25 mM Tris-HCl (pH 8), 150 mM NaCl, 5 mM MgCl$_2$, 1 mM TCEP, 0.1 mg/mL BSA for 1 h at 37°C. Data represent five independent experiments. (C, D) First derivatives of melting curves for truncated *Pv*AspRS (1.5 µM) in apo

form or after incubation at 37°C for 3 h with 10 µM ATP, 20 µM Asp, 80 µM *Ect*RNA, 10–50 µM DACM (C) or 2–10 µM Asp-DACM (D). Data are representative of three independent experiments.
(TIF)

**S7 Fig. Comparison of crystal structures of Asp-DACM-bound *Pv*AspRS with apo *Pv*AspRS.** (A) Overlay of chain A of apo *Pv*AspRS and Asp-DACM-bound *Pv*AspRS showing the flipping loop and the motif II loop. (B) Ribbon representation of the flipping loop of chain B of apo *Pv*AspRS. The SEGG loop interacts with a conserved residue Q371, as well as N356, S350 and A357. (C) B-factor analysis of the chain A and B flipping loops of different *Pv*AspRS structures. The x-axis shows residue number.
(TIF)

**S8 Fig. Crystal structure of Asp-AMP-bound *Pv*AspRS and comparison with the apo and Asp-DACM-bound structures.** (A) $2F_o$-$F_c$ maps contoured at 2 σ (mesh surface) showing electron density supporting the position of Asp-AMP bound to chains A and B. (B) Ligplots of Asp-AMP active site interfaces for the A and B chains. Hydrogen bonds and salt bridges are depicted with dashed (green) lines. Other interactions between protein and ligand are indicated by red arcs. (C) Overlay of *Pv*AspRS (Apo), *Pv*AspRS (Asp-DACM) and *Pv*AspRS (Asp-AMP) showing the flipping loop and the motif II loop. Chain A (left), Chain B (right). The sidechain of residue E398 of apo *Pv*AspRS Chain A is not built due to insufficient density.
(TIF)

**S1 Table. Thermal stabilization of recombinant *Pf*TyrRS, native length *Pf, Pv, Hs*AspRS and truncated *Pv*AspRS by nucleoside sulfamates.** *N.D. = Not determined due to incomplete transition.
(PDF)

**S2 Table. Protein–ligand docking scores determined using the Surflex fragment matching strategy.**
(PDF)

**S3 Table. X-ray diffraction data collection and refinement statistics for truncated apo *Pv*AspRS and in complex with ligands Asp-DACM, Asp-AMP and Asp-AMS.**
(PDF)

**S1 Text. Chemistry methods.**
(DOCX)

## Acknowledgments

AMS was kindly provided by Steven Langston, Drug Discovery Sciences, Takeda Pharmaceuticals International Company, Cambridge, United States. We would like to thank the University of Melbourne Bio21 Molecular Science and Biotechnology Institute's Melbourne Mass Spectrometry and Proteomics Facility, the Bio21-WEHI crystallization facility, and the Melbourne Protein Characterization Facility, including Roxanne Smith and Yee-Foong Mok, for technical support. This research was partly undertaken at the Australian Synchrotron, part of the Australian Nuclear Science and Technology Organization, and made use of the ACRF Detector on the MX2 beamline. We thank the beamline staff for their assistance. We are grateful for beamtime at the Diamond Light Source (DLS) and the staff of beamline I03 for data collection (BAG application mx28534). We are also grateful for beamtime at the SOLEIL beamlines PROXIMA 1 and 2A, and the staff of beamlines for their aid in preliminary data collection.

## Author contributions

**Conceptualization:** Nutpakal Ketprasit, Chia-Wei Tai, Vivek Kumar Sharma, Yogavel Manickam, Yogesh Khandokar, Con Dogovski, Santosh Panjikar, Sally-Ann Poulsen, Michael D.W. Griffin, Amit Sharma, Leann Tilley, Stanley C Xie.

**Formal analysis:** Nutpakal Ketprasit, Chia-Wei Tai, Vivek Kumar Sharma, Yogavel Manickam, Yogesh Khandokar, Xi Ye, Con Dogovski, David H Hilko, Craig J. Morton, Bagale Siddharam, Pushpangadan Indira Pradeepkumar, Santosh Panjikar, Sally-Ann Poulsen, Michael D.W. Griffin, Amit Sharma, Leann Tilley, Stanley C Xie.

**Funding acquisition:** Sally-Ann Poulsen, Michael D.W. Griffin, Amit Sharma, Leann Tilley, Stanley C Xie.

**Investigation:** Nutpakal Ketprasit, Chia-Wei Tai, Vivek Kumar Sharma, Yogavel Manickam, Yogesh Khandokar, Xi Ye, Con Dogovski, David H Hilko, Craig J. Morton, Anne-Sophie C Braun, Michael G. Leeming, Bagale Siddharam, Gerald J Shami, Pushpangadan Indira Pradeepkumar, Michael D.W. Griffin, Stanley C Xie.

**Supervision:** Michael D.W. Griffin, Amit Sharma, Leann Tilley, Stanley C Xie.

**Writing – original draft:** Nutpakal Ketprasit, Chia-Wei Tai, Vivek Kumar Sharma, Yogavel Manickam, Yogesh Khandokar, David H Hilko, Bagale Siddharam, Pushpangadan Indira Pradeepkumar, Santosh Panjikar, Sally-Ann Poulsen, Michael D.W. Griffin, Amit Sharma, Leann Tilley, Stanley C Xie.

**Writing – review & editing:** Nutpakal Ketprasit, Chia-Wei Tai, Vivek Kumar Sharma, Yogavel Manickam, Yogesh Khandokar, Xi Ye, Con Dogovski, David H Hilko, Craig J. Morton, Anne-Sophie C Braun, Michael G. Leeming, Bagale Siddharam, Gerald J Shami, Pushpangadan Indira Pradeepkumar, Santosh Panjikar, Sally-Ann Poulsen, Michael D.W. Griffin, Amit Sharma, Leann Tilley, Stanley C Xie.

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
