## [Decision Letter · Decision Letter 0]

PPATHOGENS-D-25-00678

Natural product-mediated reaction hijacking mechanism validates Plasmodium aspartyl-tRNA synthetase as an antimalarial drug target

PLOS Pathogens

Dear Dr. Xie,

Thank you for submitting your manuscript to PLOS Pathogens. After careful consideration, we feel that it has merit but does not fully meet PLOS Pathogens's publication criteria as it currently stands. Therefore, we invite you to submit a revised version of the manuscript that addresses the points raised during the review process.

Please submit your revised manuscript within 30 days Jul 04 2025 11:59PM. If you will need more time than this to complete your revisions, please reply to this message or contact the journal office at plospathogens@plos.org. Please include the following items when submitting your revised manuscript:

We look forward to receiving your revised manuscript.

Kind regards,

Maria Belen Cassera, PhD

Guest Editor

PLOS Pathogens

Tracey Lamb

Section Editor

PLOS Pathogens

Sumita Bhaduri-McIntosh

Editor-in-Chief

PLOS Pathogens

orcid.org/0000-0003-2946-9497

Michael Malim

Editor-in-Chief

PLOS Pathogens

orcid.org/0000-0002-7699-2064

**Additional Editor Comments :**

This is a very elegant, well-designed and written that sets the stage for future drug development of novel antimalarials. The study's findings are thorough, detailing DACM's mechanism and structural data, though its impact is slightly reduced by existing research. Reviewers have raised some minor points that can be addressed such as regarding IC50 comparisons between P. falciparum and HepG2 cells due to different treatment durations. The authors should provide IC50 for the same treatment times or justify the comparison. Another minor concern includes the need for more detailed investigation in the 0-10 nM range of inhibition. In addition, the discussion could be more concise and should compare DACM to other inhibitors, explore its sub-toxic utility, and clarify its main targets and potential modifications to reduce toxicity.

**Journal Requirements:**

1) Please provide an Author Summary. This should appear in your manuscript between the Abstract (if applicable) and the Introduction, and should be 150-200 words long. The aim should be to make your findings accessible to a wide audience that includes both scientists and non-scientists. Sample summaries can be found on our website under Submission Guidelines:

https://journals.plos.org/plospathogens/s/submission-guidelines#loc-parts-of-a-submission

- ® on page: 19

- TM on pages: 14, 15, 16, 17, and 18.

4) We have noticed that you have uploaded Supporting Information files, but you have not included a complete list of legends. Please add a full list of legends for your Supporting Information files after the references list.

5) We notice that your supplementary Tables are included in the manuscript file. Please remove them and upload them with the file type 'Supporting Information'. Please ensure that each Supporting Information file has a legend listed in the manuscript after the references list.

6) Thank you for stating "The following structures have been deposited in the PDB: PvAspRS-apo (9M5M); PvAspRS:Asp-AMP (9M5N), PvAspRS:Asp-AMS (9M5O); PvAspRS:Asp-DACM (9NPJ)." Please note that, though access restrictions are acceptable now, your entire minimal dataset will need to be made freely accessible if your manuscript is accepted for publication. This policy applies to all data except where public deposition would breach compliance with the protocol approved by your research ethics board. If you are unable to adhere to our open data policy, please kindly revise your statement to explain your reasoning and we will seek the editor's input on an exemption.

7) Please ensure that the funders and grant numbers match between the Financial Disclosure field and the Funding Information tab in your submission form. Note that the funders must be provided in the same order in both places as well. Currently, " AINSE Ltd. Postgraduate Research Award (PGRA)" is missing from the Funding Information tab. In addition, this grant "DP220102618" is missing from the Financial Disclosure field.

**Reviewers' Comments:**

Reviewer's Responses to Questions

**Part I - Summary**

Reviewer #1: This manuscript examines a bacterial natural product, DACM, as an amino acyl tRNA synthetase inhibitor against Plasmodium spp. parasites. This study builds on previous work from this team identifying substrate mimetic compounds that inhibit parasite aaRSs through initial reaction with target enzymes and subsequent failure, which permanently poisons the enzyme as an uncleavable adduct – thus “reaction hijacking.” The investigators hypothesize that the antimalarial activity of DACM is likewise through a similar mechanism. Promising features of DACM include an established synthesis route and low nM potency versus wild-type parasites; mammalian cell toxicity is reduced but still potent, 47nM). On target mechanism is supported by eIF2alpha phosphorylation and reduced protein synthesis, both expected outcomes of aaRS inhibition. Targeted mass spectrometry was used to confirm formation of inhibitor-enzyme adducts in drug-treated cells, but only for a select number of aaRSs. Lack of interaction with DACM and TyrRS is additionally supported by DSF. Molecular interaction between one of the cellular target aaRSs, AspRS, was further explored, with DSF demonstrating direct interaction between DACM and Pf and PvAspRS. Finally, the authors solve the structure of apo- and DACM-bound PvAspRS (compared to ACS and AMP-bound), identifying specific differences that enable some species selectivity. Overall, this is a comprehensive, well designed, and well written manuscript. While the toxicity of DACM is substantial and likely to limit onwards development (as noted by the authors), this study provides important proof-of-concept of aaRS inhibition that can have some species and aaRS selectivity, and the structures may be used in rational design of novel inhibitors with improved selectivity/reduced toxicity. A few minor queries are noted below:

Reviewer #2: This study examines the antimalarial activity of dealanylascamycin (DACM), a natural product from Streptomyces, and a synthetic analogue, AMS. Both compounds are nucleoside sulfamates and inhibit Plasmodium falciparum growth at levels similar to dihydroartemisinin. The authors provide data to elucidate the mechanism of DACM antimicrobial activity, showing that it works by blocking protein synthesis through eIF2α phosphorylation and targets multiple aminoacyl tRNA synthetases, including aspartyl tRNA synthetase (AspRS). They further show that DACM and AMS form stable inhibitory conjugates with aspartic acid, affecting both P. falciparum and P. vivax AspRS, but have less effect on the human enzyme. X-ray crystallography shows structural differences between parasite and human AspRS, which may explain this selectivity.

Overall, the paper is well-written, and I enjoyed reading it. This is a very thorough study that explains the mechanism of these analogs and provides compelling structural data to explain differences in selectivity. The impact is reduced slightly by the large body of previous work in this area; however, new and important results are provided herein.

Reviewer #3: Ketprasit et al present a well put together study on the inhibition of Plasmodium Aspartyl-tRNA synthetase by a proposed antimalarial drug, DACM that has previously been used as an antibacterial and herbicide. The mode of action and target of DACM was not previously characterized in detail.

The authors synthesize a naturally occurring compound, DACm (produced by a Streptomyces species), and test DACM in P, falciparum growth assays and show a strong inhibitory effect on growth in the nM range. In comparison, HepG2 cells are 10 times less sensitive , and show now inhibition at concentrations that are inhibitory to P. falciparum. This is strong evidence that DACM could be a useful antimalarial drug. They follow up with well controlled puromycylation assays and identified reduced protein translation. Next they identified several synthetases as potential targets for DACM, and followed up on AspRS as one other the main targets. They in detail characterized the binding of DACM to Plasmodium AspRS, including DACM bound structure that shows a slight rearrangement of the protein due to Asp-DACM binding.They compared the structure to Asp-AMS bound AspRS and found similar interactions. Overall this is a well thought through and carried out study.

**Part II – Major Issues: Key Experiments Required for Acceptance**

Reviewer #1: none

Reviewer #2: 1. My primary concern lies with comparisons of IC50 between P. falciparum and HepG2 cells. For P. falciparum, the IC50_72h is reported; however, for HepG2, IC50_48h is reported. The authors use these values to claim that DACM is ten-fold less toxic to the HepG2 cells. It is not immediately clear to me how these IC50 values can be used in a one-to-one comparison, given the differences in treatment time. A previous study, cited by the authors (i.e., ref 17) uses IC50_72h for both P. falciparum and HepG2 which seems much more reasonable. This is a very important point, because the authors wish to tie toxicity to their structural/biochemical work. The authors should provide IC50 values for cells treated for the same time, or provide a reasonable explanation for why these values can be compared as is.

Reviewer #3: Minor concerns:

Fig 1B - this is great data, and it would be useful to in more detail investigate the 0-10 nM range to identify the exact range of inhibition. I would suggest to have a few more datapoints for Plasmodium between no and full viability.

Fig 1C - please add the blot for eif1a (non-phospho) to show if the relative phosphorylation if up/ eif2a abundance is changed.

Discussion:

The discussion could be a more concise, as it mainly repeats the results. How does DACM compare to other translation inhibitors and other antimalarials? Could DACM still be useful at sub-toxic concentrations? What were the DACM experiments in mice and what does your data contribute to those studies now that you know the target of DACM?

Do you think AspRS is the main target for DACM? Or does it target the other synthetases you identified equally? Would you speculate a similar mode of action? How do you think DACM could be further modified to reduce toxicity?

**Part III – Minor Issues: Editorial and Data Presentation Modifications**

Reviewer #1: 1. Fig 1 – it is important for antimalarials to maintain activity against clinically important drug-resistance mechanism. Please provide IC50s against well characterized chloroquine-resistant and K13 mutant (artemisinin reduced sensitivity) parasite lines.

2. Source of human erythrocytes should be indicated (and relevant ethical approvals noted if relevant).

3. Source of P. falciparum 3D7 parasites should be stated and growth conditions for culture should be provided (or cited) in methods.

4. For ATP consumption assay, please confirm that assay conditions were in the linear range with respect to time and enzyme concentrations.

Trivial edits:

1. First paragraph of methods, I believe the heading should read “inhibition of the growth [of] P. falciparum cultures”

Reviewer #2: 2. The use of “Figure S1” and “Supplementary Figure 1” in the SI is very confusing. It took me a quite a long time to find the correct data referenced in the article.

Reviewer #3: (No Response)

PLOS authors have the option to publish the peer review history of their article (what does this mean? ). If published, this will include your full peer review and any attached files.

**Do you want your identity to be public for this peer review?** For information about this choice, including consent withdrawal, please see our Privacy Policy .

Reviewer #1: No

Reviewer #2: No

Reviewer #3: No

**Figure resubmission:**
---

## [Decision Letter · Decision Letter 1]

Dear Dr Xie,

We are pleased to inform you that your manuscript 'Natural product-mediated reaction hijacking mechanism validates Plasmodium aspartyl-tRNA synthetase as an antimalarial drug target' has been provisionally accepted for publication in PLOS Pathogens.

Best regards,

Maria Belen Cassera, PhD

Guest Editor

PLOS Pathogens

Tracey Lamb

Section Editor

PLOS Pathogens

Sumita Bhaduri-McIntosh

Editor-in-Chief

PLOS Pathogens

orcid.org/0000-0003-2946-9497

Michael Malim

Editor-in-Chief

PLOS Pathogens

orcid.org/0000-0002-7699-2064

Reviewer Comments (if any, and for reference):

Reviewer's Responses to Questions

**Part I - Summary**

Reviewer #1: The authors have responded well to my critiques. I would prefer to have IC50s vs. drug-resistant parasite lines included in this manuscript, but it was always a minor concern given their prior data.

Reviewer #2: The authors have sufficiently addressed my concern.

Reviewer #3: This is a revision.

**Part II – Major Issues: Key Experiments Required for Acceptance**

Reviewer #1: none

Reviewer #2: (No Response)

Reviewer #3: All concerns have been addressed, or a sufficient justification was provided why additional experiments are not feasible.

**Part III – Minor Issues: Editorial and Data Presentation Modifications**

Reviewer #1: none

Reviewer #2: (No Response)

Reviewer #3: All concerns have been addressed, or a sufficient justification was provided why additional experiments are not feasible.

PLOS authors have the option to publish the peer review history of their article (what does this mean? ). If published, this will include your full peer review and any attached files.

**Do you want your identity to be public for this peer review?** For information about this choice, including consent withdrawal, please see our Privacy Policy .

Reviewer #1: No

Reviewer #2: No

Reviewer #3: No

---

## [Editor Report · Acceptance letter]

Dear Dr Xie,

We are delighted to inform you that your manuscript, "Natural product-mediated reaction hijacking mechanism validates Plasmodium aspartyl-tRNA synthetase as an antimalarial drug target," has been formally accepted for publication in PLOS Pathogens.

Best regards,

Sumita Bhaduri-McIntosh

Editor-in-Chief

PLOS Pathogens

orcid.org/0000-0003-2946-9497

Michael Malim

Editor-in-Chief

PLOS Pathogens

orcid.org/0000-0002-7699-2064